# Inflammatory and infectious upper respiratory diseases associate with 41 genomic loci and type 2 inflammation

Elmo C. Saarentaus [1,2], Juha Karjalainen[1,3,4], Joel T. Rämö [1,5], Tuomo Kiiskinen [1,6], Aki S. Havulinna[1,6], Juha Mehtonen [1], Heidi Hautakangas [1], Sanni Ruotsalainen [1], Max Tamlander [1], Nina Mars[1,7], FINNGEN*, Sanna Toppila-Salmi[8], Matti Pirinen [1,9,10], Mitja Kurki[1,3,4], Samuli Ripatti [1,7,10], Mark Daly[1,4,7], Tuula Palotie[11,12], Antti Mäkitie [2] & Aarno Palotie [1,3,13] ✉

Inflammatory and infectious upper respiratory diseases (ICD-10: J30-J39), such as diseases of the sinonasal tract, pharynx and larynx, are growing health problems yet their genomic similarity is not known. We analyze genome-wide association to eight upper respiratory diseases (61,195 cases) among 260,405 FinnGen participants, meta-analyzing diseases in four groups based on an underlying genetic correlation structure. Aiming to understand which genetic loci contribute to susceptibility to upper respiratory diseases in general and its subtypes, we detect 41 independent genome-wide significant loci, distinguishing impact on sinonasal or pharyngeal diseases, or both. Fine-mapping implicated non-synonymous variants in nine genes, including three linked to immune-related diseases. Phenome-wide analysis implicated asthma and atopic dermatitis at sinonasal disease loci, and inflammatory bowel diseases and other immune-mediated disorders at pharyngeal disease loci. Upper respiratory diseases also genetically correlated with autoimmune diseases such as rheumatoid arthritis, autoimmune hypothyroidism, and psoriasis. Finally, we associated separate gene pathways in sinonasal and pharyngeal diseases that both contribute to type 2 immunological reaction. We show shared heritability among upper respiratory diseases that extends to several immune-mediated diseases with diverse mechanisms, such as type 2 high inflammation.

Inflammatory and infectious upper respiratory diseases (IURD) affect the sinonasal tract, pharynx, and larynx, and include diseases such as chronic tonsillitis, allergic rhinitis, and chronic rhinosinusitis (CRS). They lead to increased morbidity[1,2] and costs[3], and to the highest public health burden in the world[4] by serving as the main route of infection to the body, and by their connection to non-communicable diseases, such as asthma[5–7], autoimmune diseases[8,9], cardiovascular diseases[10], and obesity[11]. Genetic predisposition[12–15] together with environmental megatrends such as the COVID-19 pandemic[16–18], Western lifestyle[19], urbanization[20,21], global warming[22], and dysbiosis[23,24] influence the burden of IURDs. IURDs often co-exist[25–28], and they have shown overlapping mechanisms[5,6,29,30]. Understanding the genetic (dis)

A full list of affiliations appears at the end of the paper. *A list of authors and their affiliations appears at the end of the paper.
✉e-mail: aarno.palotie@helsinki.fi

similarities behind IURDs can remarkably improve preventive actions and therapies, and reduce the burden of IURDs and related diseases[2,31].

IURDs are characterized by an etiology related to recurrent infections and dysbiosis[20,21,24] leading to chronic and treatment-resistant diseases[32] with acute and even life-threatening exacerbations. IURDs involve inflammation in the nasal cavity, such as vasomotor and allergic rhinitis (VAR), both characterized by hyperresponsiveness to stimuli[33]; non-specific chronic rhinitis, nasopharyngitis and pharyngitis (CRNP) and nasal septal deviation (NSD); and in the adjoining paranasal sinuses, such as CRS with or without nasal polyps (NP)[6]. Allergic rhinitis (AR) is a part of an allergic disease entity involving allergic asthma, atopic dermatitis, allergic conjunctivitis and food allergy[27,34,35]. IURDs also encompass other diseases of the pharynx such as chronic laryngitis and laryngotracheitis (CLT), chronic diseases of tonsils and adenoids (CDTA), and peritonsillar abscess (PA). Previous genetic studies of non-allergic IURDs and related immune responses have largely focused on rare variants[36] and the *HLA* region[37,38]. IURD-related GWAS have been reported of CRS and NP[39], tonsillectomy and childhood ear infections[40,41], cold sores, mononucleosis, strep throat, pneumonia and myringotomy[40], and of infective diseases caused by specific airway-related microbes such as pneumococcus[42] and staphylococcus aureus[43]. The common variant burden of allergic diseases such as AR have been more extensively studied[41,44–48]. However, no prior research has analyzed shared genetic contributions of IURDs.

The FinnGen study is a large biobank study including both genetic and lifelong health record data from all participants, thus allowing the investigation of potentially shared and distinct genetic landscape associated to IURDs. This provides an opportunity both for GWASs as well as for cross-disease analyses to better understand potential shared genetic contributors. We aimed to study genetic predispositions to recurrent, chronic and complicated IURDs. We hypothesized that, on one hand, shared genetic variants contribute to IURD susceptibility in general, and some variants contribute more to distinct IURD phenotypes. To test this hypothesis, we analyzed genome-wide association of IURD cases in the FinnGen study (release 6 Aug 2020), a nation-wide collection of genotyped samples from Finnish individuals. Our study sample included 260,405 individuals of all ages, where we focused on cases of specialist-diagnosed IURDs ($n = 61,197$), including their more specific diagnosis. We tested the genetic associations across IURDs to highlight shared and distinct genetic contributions among IURDs. Finally, we compared the genome-wide association of IURDs and phenotypes to other anatomically related and systemic immunological disorders (such as chronic periodontitis; CP) linked with the same genetic loci.

## Results

### Genome-wide association of IURD

We performed genome-wide association analysis of all IURD cases ($n = 61,197$, ranging from 2623 to 29,135 per phenotype) in FinnGen (Table 1). We genotyped and imputed 16,355,289 single-nucleotide genetic variants in 260,405 Finnish individuals of all ages. We used a logistic mixed model with the SAIGE software[49] (see Methods) to detect genome-wide association between 61,197 cases of different IURD diagnoses (Table 1, Supplementary Figure 1) using the same 199,208 controls for all IURDs, and set as covariates age, genetic sex, principal components (PCs) 1–10, and genotyping batch. In addition to the main phenotypes linked to the upper respiratory tract, we also analyzed genome-wide association to two oral inflammatory diseases that have been associated[50–52] with IURDs: diseases of pulp and periapical tissues (DPPT; ICD-10 K04, 48,687 cases vs 211,718 controls) and CP (ICD-10 K05.30-.31, 14,631 cases vs 245,774 controls). We set the level of multiple testing significance (MTS) at $p < 5e{-}09$ for ten independent phenotypes. Using the FinnGen study sample the eight different IURD GWASs detected 907 MTS variant associations in 25 independent loci in total (Supplementary Data 1).

### IURD shared heritability

To explore the shared genetic risk landscape for different upper respiratory diseases, we analyzed the potentially shared heritability between different diagnostic entities. Thirteen of the 25 loci showed similar impact among different IURDs (Fig. 1A). We used hierarchical clustering of lead variant effect estimates to group loci and phenotypes. The variant effects largely correlated among VAR and CRS as one group, and the two tonsillar diseases, CDTA and PA, as another. Hierarchic clustering also distinguished broadly shared impact among VAR, CRS, and NP in four loci (2q12.1, 5q22.1, 9p24.1, 10p14b). The 2q33.3 locus was broadly associated with upper respiratory diseases with a concordant impact among CDTA, VAR, CRS, and NP. In total, 13 of 24 non-*HLA* IURD loci had a co-directional association ($p < 0.00027$) with at least one other IURD phenotype in line with the hypothesis for a shared genetic background.

We next used LD Score regression[53] based genome-wide correlation analysis to explore the shared genetic background of IURDs. This distinguished three IURD phenotype clusters from the GWAS results (Fig. 1B). A high genetic correlation ($r_g > 75\%$) distinguished two clusters: (I) VAR, CRS, NP, and NSD ($r_g \geq 78\%$); (II) CDTA and PA ($r_g = 79\%$), in line with results from the hierarchical cluster analysis. Using a threshold of $r_g > 90\%$ further distinguished a genetically linked subgroup of known comorbid disorders[26]: (III) VAR, CRS, NP. We denoted these IURD groups as "sinonasal diseases" (I), "pharyngeal diseases" (II), and "chronic inflammatory sinonasal diseases" [CISDs, (III), Fig. 2].

**Table 1 | Description of genome-wide association studies**

| Abbr. | Phenotype | ICD-10 | Cases | $\lambda_{GC}$ | Loci | CS |
|---|---|---|---|---|---|---|
| VAR | Vasomotor and allergic rhinitis | J30 | 8975 | 1.0772 | 3 | 3 |
| CRNP | Chronic rhinitis, nasopharyngitis, and pharyngitis | J31 | 6518 | 1.0354 | 0 | 0 |
| CRS | Chronic rhinosinusitis | J32 | 10,435 | 1.0864 | 4 | 3 |
| NP | Nasal polyps | J33 | 3919 | 1.0618 | 9 | 9 |
| NSD | Nasal septal deviation | J34.2 | 7716 | 1.0584 | 0 | 0 |
| CDTA | Chronic diseases of tonsils and adenoids | J35 | 29,135 | 1.1908 | 14 | 14 |
| PA | Peritonsillar abscess | J36 | 4863 | 1.0527 | 3 | 3 |
| CLT | Chronic laryngitis and laryngotracheitais | J37 | 2623 | 1.0204 | 0 | 0 |
| DPPT | Diseases of pulp and periapical tissues | K04 | 48,687 | 1.1175 | 1 | 1 |
| CP | Chronic periodontitis | K05.30-1 | 14,631 | 1.0610 | 0 | 0 |

The cases were identified using registry data from hospitals and specialized out-patient clinics. *Abbr.* Abbreviation for phenotype. The same set of controls ($n = 199,208$) was used in all IURD GWASs (ICD-10 category J3). Control counts were 211,718 for DPPT and 245,774 for CP. $\lambda_{GC}$ is the genomic inflation factor. 'Loci' is the number of multiple testing significant (MTS; $p < 5e{-}9$) loci (incl HLA). *P* values were calculated using upper tail chi-square testing (one degree of freedom). CS is the number of credible sets from fine-mapping with at least one MTS SNP (the HLA region was not fine-mapped). For details see Supplementary Fig. 1 and Supplementary Data 1.

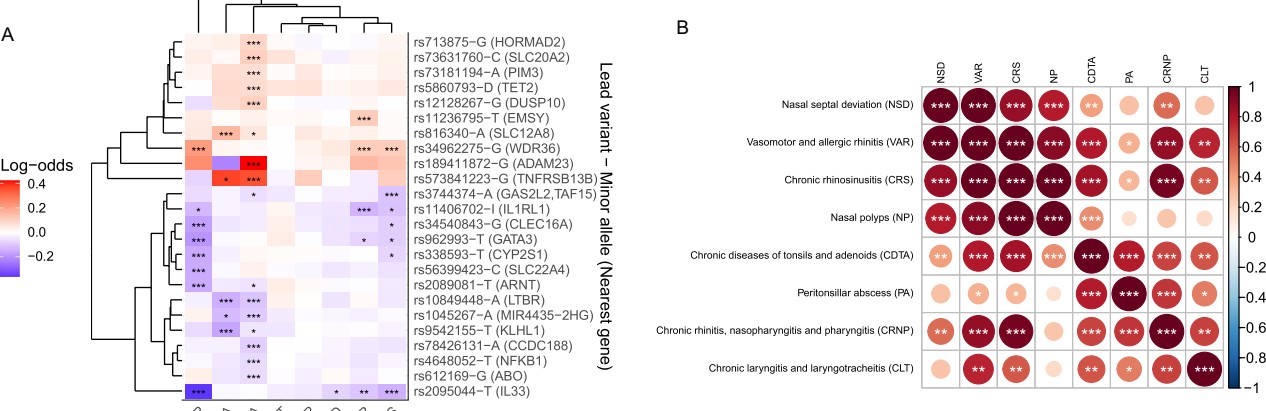

**Fig. 1 | Shared heritability among inflammatory and infectious upper respiratory diseases (IURDs). A** (left): effect sizes of lead variants of 24 non-*HLA* loci across IURD phenotypes. Red indicates a positive and blue a negative effect size estimate (in log-odds) using logistic regression (Methods). *P* values were calculated using upper tail chi-square testing (one degree of freedom) from a t-statistic under a normal approximation. Variants and phenotypes are ordered according to hierarchical clustering (Methods). The clusters show shared genetic heritability for variant clusters between recognized phenotype groups of sinonasal (NSD, VAR, CRS, NP) and pharyngeal diseases (CDTA, PA). *$p < 0.00027$, **$p < 5e{-}8$, ***$p < 5e{-}9$. **B** (right): genetic correlation of IURDs distinguishing vasomotor and allergic rhinitis (VAR), chronic rhinosinusitis (CRS), and nasal polyposis (NP) as a near-completely genetically correlated cluster. The color of the circle indicates genetic correlation with red indicating positive correlation and blue indicating negative correlation. *P* values were calculated using upper tail chi-square testing (one degree of freedom) from a z statistic. *$p < 0.05$, **$p < 0.005$, ***$p < 8.4e{-}5$.

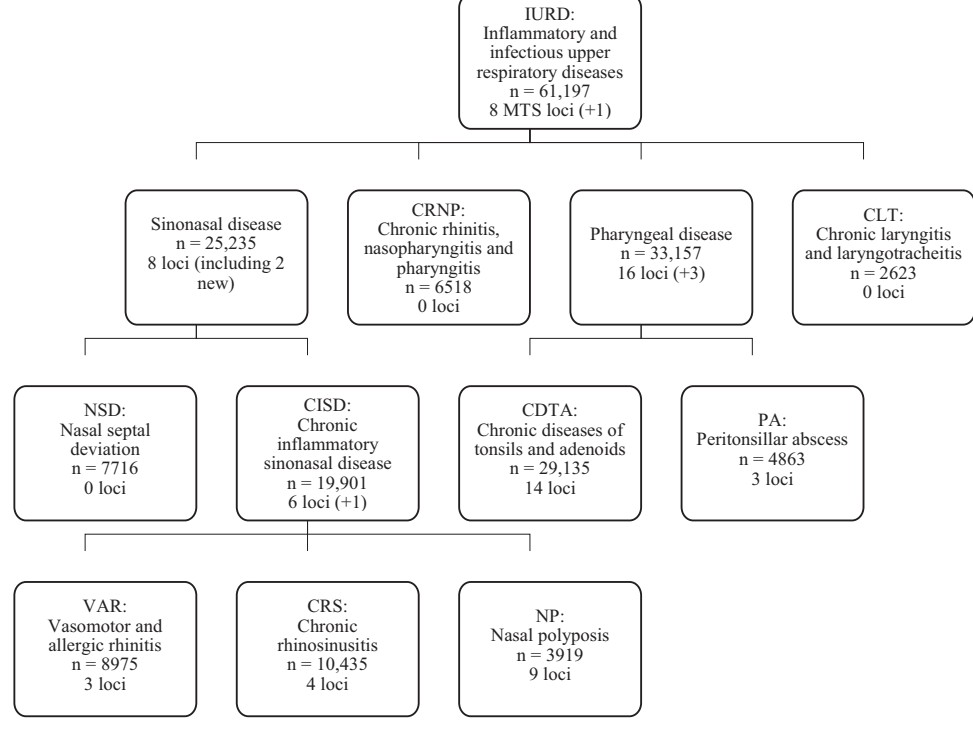

**Fig. 2 | The IURD phenotype structure, based on genetic correlation between phenotypes.** The boxes represent a GWAS of a IURD phenotype or group, stating the name, case count, and a number of MTS loci, indicating in parenthesis the MTS loci that were not detected in any directly preceding ("child") GWAS. E.g., there were two loci in sinonasal disease GWAS that had not been detected in NSD, CISD, VAR, CRS, or NP GWASs. The hierarchical structure shows the phenotypes included in the parent phenotype. The IURD GWAS also included ICD codes J38 and J39 (not depicted); for these, no separate GWAS was performed. Sinonasal disease phenotypes (NSD, VAR, CRS, and NP) had a genetic correlation 78% or higher, as estimated using LD Score regression. Pharyngeal diseases (CDTA and PA) had a genetic correlation of 79%. The observed genetic correlations between chronic inflammatory sinonasal diseases (VAR, CRS, and NP) were 90% or higher.

The CRNP phenotype had high genetic correlation with both pharyngeal diseases ($r_g \geq 67\%$) and VAR and CRS ($r_g \geq 87\%$), but not NP ($p = 0.051$).

Using the same GWAS pipeline as described above, cross-trait analysis using these IURD clusters identified six additional MTS loci. We performed cross-trait GWASs of sinonasal diseases ($n = 25{,}235$, Supplementary Table 1, Supplementary Figure 2A), pharyngeal diseases ($n = 33{,}157$, Supplementary Data 2, Supplementary Figure 2B), and CISD ($n = 19{,}901$, Supplementary Table 2, Supplementary Figure 2C). In addition, we performed a GWAS of cases with any IURD ($n = 61{,}197$, Supplementary Data 3, Supplementary Figure 3). The genome-wide significant (GWS, $p < 5e{-}08$) VAR-associated locus 9q33.3

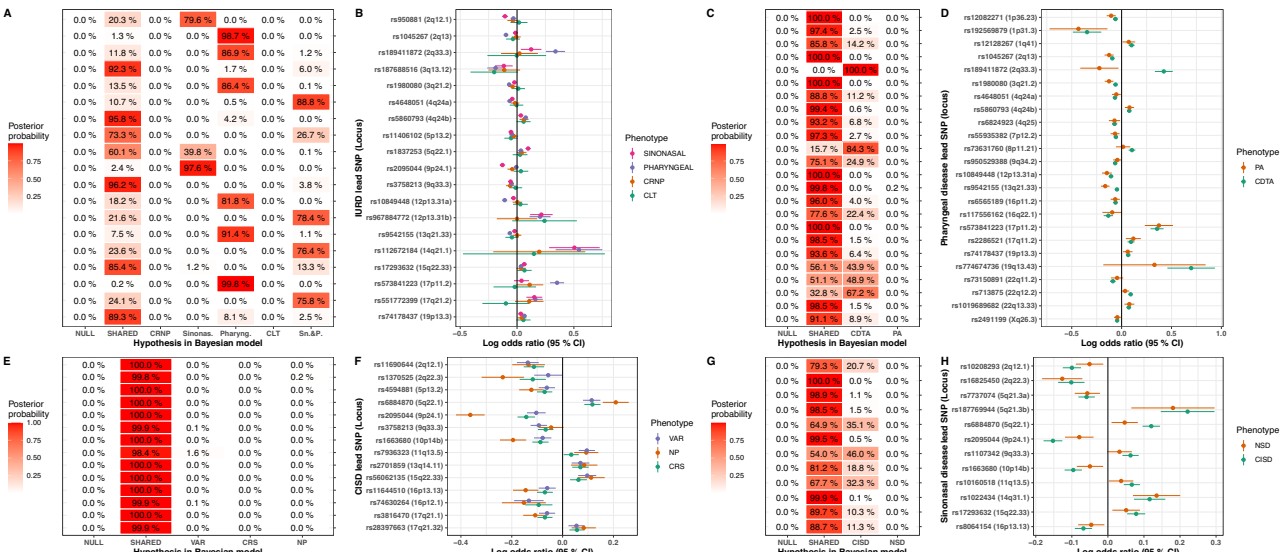

**Fig. 3 | Shared impact between phenotypes for cross-trait analysis lead variants. A** (upper left): Bayesian posterior probabilities of hypothetical models displayed on the x-axis for IURD lead variants (y-axis; locus in parenthesis). Models correspond to NULL = null model; SHARED = Fixed or correlated effect model across phenotypes (CRNP, sinonasal disease, pharyngeal disease and CLT); CRNP = CRNP only; Sinonas. = sinonasal disease only; Pharyng. = pharyngeal disease only; CLT = CLT only; Sn.&P. = Sinonasal AND pharyngeal disease (fixed-effect). A model with PIP > 70% was considered likely. **B** (upper left): Odds ratio point estimates of IURD lead SNPs for phenotypes, with 95% confidence intervals [n (sinonasal) = 25,235 cases, n (pharyngeal) = 33,157 cases, n (CRNP) = 6518 cases, n (CLT) = 2623 cases]. **C** (upper right): Bayesian posterior probabilities of hypothetical models displayed on the x axis for pharyngeal disease lead SNP (y axis). Models correspond to NULL = null model; CORR = Fixed or correlated effect across phenotypes (CDTA and PA); CDTA = CDTA only; PA = PA only. **D** (upper right): Odds ratio point estimates of pharyngeal lead variants for CDTA and PA, with 95% confidence intervals [n (PA) = 4863 cases; n (CDTA) = 29,135 cases]. **E** (lower left): Bayesian posterior probabilities of hypothetical models for each lead variants from the sinonasal disease GWAS. Models as in **A**; additionally CISD = CISD diseases only; and NSD = NSD only. **F** (lower left): Odds ratio point estimates of sinonasal disease lead variants for the two phenotypes, with 95% confidence intervals [n (NSD) = 7716 cases; n (CISD) = 19,901 cases]. **G** (lower right): Bayesian posterior probabilities of hypothetical models for each lead variants from the sinus disease GWAS. Models NULL and SHARED as in **A**; additionally VAR VAR only, CRS CRS only, NP NP only. **H** (lower right): Odds ratio point estimates of sinus disease lead variants for the three phenotypes VAR, CRS, and NP, with 95% confidence intervals [n (VAR) = 8975 cases; n (CRS) = 10,435 cases; n (NP) = 3919 cases].

(near *NEK6*) was MTS associated with all IURDs [OR = 0.95 (0.93–0.97), $p = 1.75e{-10}$]. Similarly, the GWS CDTA-associated loci 1p36.23 and 16p11.2 were MTS associated with pharyngeal diseases, and the NP-associated locus 2q22.3 near *ZEB2* was MTS associated with sinonasal diseases. The GWAS of CISD identified the 15q22.33 locus near *SMAD3* [OR = 1.08 (1.05–1.11), $p = 9.38e{-10}$], previously associated[45] with allergic disease, that was not observed in GWASs of VAR, CRS or NP. GWAS of sinonasal diseases additionally identified the locus 14q31.1 near *NRXN3* [OR = 1.13 (1.08–1.18), $p = 3.47e{-09}$], not detected by CISD or NSD GWASs and not previously implicated. The six additional loci from cross-trait analyses brought our yield to 31 IURD-associated non-*HLA* loci.

To provide further robustness of our cross-trait analyses, we ran the MultiTrait Analysis of GWAS (MTAG[54], see Methods) software. The MTAG analysis on all IURD traits supported three (near *SMAD3*, *IL7R*, and *IKZF3*) of the six loci observed in the cross-trait analysis above, providing additional confidence for these associations (Supplementary Data 4). Among the IURD traits with no MTS loci (CRNP, NSD and CLT), MTAG supported CRNP association for four loci, of which 9q33.3 near *NEK6* [OR = 0.99 (0.98–1.00)] replicated ($p = 0.0081$) in UKB (see below). NSD was associated with eight loci also detected inVAR and CRS GWASs. MTAG analysis additionally identified four GWS loci not seen in the association analyses described above. One of these four MTAG hits replicated ($p = 0.028$) in UKB: the 11q12.2 locus near *FADS2* associated with NP [OR = 0.98 (0.97–0.99), $p = 2.7e{-8}$]. Together with the 31 independent MTS loci from IURD and cross-trait GWASs, the 11q12.2 locus brought our yield to 32 genomic loci.

**IURD distinct heritability**
We established the locus-specific shared and distinct genetic impact by comparing the associations in phenotype-specific GWAS using a

Bayesian framework (see Methods) for lead variants. Briefly, this framework tests the probability of hypothesized association models for a variant using summary statistics of the GWASs being compared, taking into account the overlapping cases and controls between phenotypes[55]. The framework allowed us to evaluate the following models: the null model, where the variant explains no part of any of the phenotype variation; the fixed model, where the variant has one fixed-effect that is the same for all phenotypes; the correlated model, where the variant has a correlating effect on all phenotypes; and models where the impact is to one phenotype only.

The Bayesian framework distinguished a subdivision for the detected loci in most cases, providing evidence that some of the variant associations were more disease-specific than others. Among the 19 non-*HLA* GWS lead variants detected in IURD GWAS (Fig. 3A, B), a shared effect was supported (P(Fixed or Correlated) >75%) for five loci. Six loci were likely only impacting pharyngeal diseases; two only sinonasal diseases; and four likely both pharyngeal and sinonasal diseases. Thus 9/19 loci were considered shared between sinonasal and pharyngeal diseases, with possible effect on CL and CRNP from five loci —the remaining loci likely being more specific in their impact. Two loci remained uncertain: rs11406102 had a less clear general impact (P(Fixed or Correlated) = 73.3%), and rs1837253 impacted sinonasal diseases with an uncertain effect on other phenotypes. In a similar vein for sinonasal diseases (Fig. 3E, F), consisting of CISD and NSD, all tested models supported an impact on CISD for all variants, and possible impact on NSD for three lead variants. The pharyngeal disease analysis showed a shared impact for 20 variants, with two variants being likely CDTA-specific and three variants impacting CDTA and possibly PA (Fig. 3C, D). Strikingly, all CISD lead variants were either consistent or highly correlated in their effect among the three subphenotypes VAR, CRS, and NP (Fig. 3G, H).

**Table 2 | Inverse-variance-weighted meta-analysis of loci lead variants with similar effect in FinnGen and UKB**

| LOCUS | RSID | Nearest gene | Consequence | MAF | Phenotype | OR | 95% CI | P-value |
|---|---|---|---|---|---|---|---|---|
| 9q33.3* | rs3758213-T | NEK6 | intronic | 38.2% | VAR | 0.95 | (0.93–0.97) | 4.81E-08 |
| 11q13.5 | rs11236795-T | EMSY | intergenic | 26.2% | VAR | 0.93 | (1.06–1.14) | 6.16E-12 |
| 1q21.3* | rs2089081-T | ARNT | intronic | 44.5% | NP | 1.10 | (1.11–1.20) | 9.55E-10 |
| 2q12.1 | rs56117144-C | IL18RAP | intergenic | 28.8% | NP | 1.15 | (0.76–0.87) | 8.32E-16 |
| 2q22.3* | rs66484168-G | ZEB2 | intergenic | 8.8% | NP | 0.81 | (0.76–0.87) | 2.95E-12 |
| 5q22.1 | rs34962275-G | WDR36 | downstream | 31.3% | NP | 1.20 | (1.16–1.25) | 7.60E-28 |
| 5q31.1a | rs11738827-T | CDC42SE2 | intronic | 27.3% | NP | 0.88 | (0.85–0.92) | 7.82E-13 |
| 5q31.1b | rs56399423-C | SLC22A4 | intronic | 31.7% | NP | 0.87 | (0.83–0.90) | 4.65E-18 |
| 9p24.1 | rs2095044-T | IL33 | upstream | 23.9% | NP | 1.34 | (1.29–1.39) | 1.20E-58 |
| 10p14a | rs10905284-C | GATA3 | intronic | 42.0% | NP | 0.88 | (0.85–0p.91) | 2.44E-15 |
| 10p14b | rs962993-T | GATA3 | regulatory | 30.3% | NP | 0.83 | (0.80–0.86) | 2.58E-28 |
| 12q13.2 | rs705702-G | RAB5B | upstream | 30.3% | NP | 1.15 | (1.10–1.19) | 5.24E-16 |
| 16p13.13 | rs34540843-G | CLEC16A | intronic | 21.2% | NP | 0.86 | (0.83–0.90) | 4.90E-14 |
| 19q13.2 | rs338593-T | CYP2S1 | intronic | 42.8% | NP | 0.87 | (0.84–0.90) | 2.25E-18 |
| 1p36.23 | rs12082271-T | SLC45A1 | upstream | 30.2% | CDTA | 0.94 | (0.91–0.96) | 2.65E-09 |
| 2p13.2* | rs35668054-T | DYSF | regulatory | 9.5% | CDTA | 1.10 | (1.06–1.14) | 2.30E-08 |
| 2q33.3 | rs189411872-G | ADAM23 | intronic | 1.3% | CDTA | 1.53 | (1.39–1.68) | 1.13E-20 |
| 4q24a | rs4648052-T | NFKB1 | intronic | 39.9% | CDTA | 0.94 | (0.91–0.96) | 1.99E-11 |
| 4q24b | rs5860793-D | TET2 | intergenic | 28.4% | CDTA | 0.93 | (0.90–0.95) | 1.60E-12 |
| 8p11.21* | rs73631760-C | SLC20A2 | intronic | 9.4% | CDTA | 1.11 | (1.07–1.15) | 1.65E-09 |
| 9q34.2 | rs612169-G | ABO | intronic | 43.6% | CDTA | 1.07 | (1.04–1.09) | 6.45E-11 |
| 12p13.31 | rs10849448-A | LTBR | 5′ UTR | 24.5% | CDTA | 1.11 | (1.08–1.14) | 3.77E-19 |
| 19p13.3 | rs74178437-G | ZBTB7A | intronic | 26.3% | CDTA | 0.94 | (0.91–0.96) | 3.83E-09 |
| 22q12.2 | rs713875-G | HORMAD2 | intronic | 47.9% | CDTA | 1.09 | (1.07–1.12) | 2.40E-20 |
| 3q12.3* | rs1456200-A | NFKBIZ | upstream | 37.5% | PA | 1.13 | (1.08–1.17) | 8.69E-09 |
| 3q21.2 | rs1980080-C | SLC12A8 | intronic | 34.7% | PA | 1.15 | (1.09–1.20) | 8.57E-11 |
| 13q21.33 | rs9542155-T | KLHL1 | intronic | 35.6% | PA | 1.16 | (1.11–1.21) | 4.11E-14 |
| 15q22.33 | rs17293632-T | SMAD3 | intronic | 26.2% | VAR** | 1.07 | (1.05–1.10) | 2.09E-11 |
| 7p12.2* | rs55935382-A | IKZF1 | intergenic | 31.4% | CDTA** | 0.94 | (0.92–0.97) | 1.16E-08 |

*MAF* minor allele frequency in FinnGen. "Consequence" reports the most severe predicted variant impact: *5′ UTR* Untranslated region in 5′ end of gene, *non-coding* exon of non-coding gene, *regulatory* regulatory region. Odds ratios (OR) were estimated using logistic regression (Methods). *P*-values were calculated using upper tail chi-square testing (one degree of freedom) from a t-statistic under a normal approximation. 95% CI were derived using normal approximation. Loci shared among phenotypes are denoted for the phenotype with the lowest *p* value, such that, e.g., 2q12.1 was MTS associated with VAR, CRS, and NP, and is shown here for NP. *locus has no previous association with IURDs **detected from cross-trait analysis and meta-analyzed using the specific IURD with smallest *p* value in FinnGen.

## Replication and meta-analysis in other cohorts

For replication and meta-analysis, we analyzed association of all GWS non-*HLA* loci lead variants identified in the FinnGen study sample in the UK Biobank (Supplementary Table 3, Supplementary Data 1 and 5). We mapped the IURD phenotypes to corresponding UKB read codes (Methods), and meta-analyzed variants with co-directional effects between the cohorts. Meta-analysis resulted in GWS association for 29 co-directional loci (Table 2). In addition to loci significant in meta-analysis, there remained eight loci with MTS association in FinnGen (Table 3). We also observed three loci with a co-directional impact ($p < 0.05$) in UKB that were considered replicated despite not reaching GWS in meta-analysis (Table 3). In this way, in addition to the 31 MTS loci detected in single-phenotype and cross-trait analyses (the FinnGen discovery phase), meta-analysis and replication supported ten additional associations. This brought our results to a grand total of 41 loci with robust IURD associations.

All five VAR associations replicated in UKB, including the *NEK6* locus, albeit at a significantly ($p_z = 0.0020$) milder impact. Three loci linked with CRS replicated in UKB. In addition to the eleven NP-associated loci overlapping ten previously reported loci in UKB, we replicated two NP loci at 1q21.3 (*ARNT*) and 2q22.3 (*ZEB2*) not previously reported.

Most loci linked to pharyngeal diseases showed high concordance in the UKB analysis despite significantly lower case counts. Lead variants of 13 of the 19 CDTA-associated non-*HLA* loci showed co-directional and similar impact ($p_z > 0.05$) between UKB and FinnGen, and ten of the 13 loci were GWS in meta-analysis (Table 2, Supplementary Data 1). Five other MTS associated CDTA loci showed counter-directional effects in UKB despite adequate power (>70%), including the high-impact ($OR_{fg} = 1.43$) 17p11.2 locus near *TNFRSF13B*, previously associated with tonsillectomy in 23andMe[40] (Table 3). This apparent inconsistency highlights the occasional challenges in replicating findings between large biobank studies, when phenotype definitions between studies are not easily translateable. Three additional loci associated with PA were GWS in meta-analysis with UKB, with the 3q21.2 locus (*SLC12A8*) also formally replicating ($OR_{ukb} = 1.23$, $p_{ukb} = 0.00054$). Statistical power for replication was <80% for seven of the CDTA lead variants as CDTA and PA had far lower effective sample sizes in UKB compared to FinnGen (4.6% for CDTA, 13.3% for PA). An additional restriction was that the single-variant CDTA locus (rs774674736-D at 19q13.43) has not been genotyped in UKB. As only one of the seven lead variants with low power did not replicate, failure to replicate is likely linked to the non-representative case count in UKB ($n = 1180$), differences in LD structure, and Finnish-specific low-frequency variants, rather than false positives in FinnGen.

We also investigated the association of previously reported GWAS of similar traits (Supplementary Data 6). We identified as 'replicated' any previously reported locus with a similar directional

**Table 3 | Lead variants of loci with heterogeneous impact between FinnGen and UKB**

| LOCUS | RSID | Nearest gene | Consequence | MAF | FinnGen | | | | UKB | | | |
|---|---|---|---|---|---|---|---|---|---|---|---|---|
| | | | | | Phenotype | OR | 95% CI | P value | Phenotype | OR | 95% CI | P value |
| 17q12* | rs3744374-A | GAS2L2 | missense | 23.0% | CRS | 0.90 | (0.86–0.93) | 9.94E-10 | CRS | 1.01 | (0.97–1.05) | 0.62 |
| 1q41 | rs12128267-G | DUSP10 | intronic | 12.2% | CDTA | 1.11 | (1.07–1.14) | 7.38E-11 | CDTA | 0.96 | (0.83–1.11) | 0.58 |
| 2q13* | rs1045267-A | MIR4435-2HG | non-coding | 34.0% | CDTA | 1.10 | (1.07–1.13) | 4.09E-17 | CDTA | 0.99 | (0.89–1.10) | 0.84 |
| 17p11.2 | rs73841223-G | TNFRSFB13B | non-coding | 2.5% | CDTA | 1.43 | (1.33–1.53) | 2.37E-26 | CDTA | 0.77 | (0.38–1.54) | 0.46 |
| 22q1.2* | rs78426131-A | CCDC188 | regulatory | 13.6% | CDTA | 0.92 | (0.88–0.95) | 2.39E-09 | CDTA | 1.01 | (0.90–1.13) | 0.83 |
| 22q13.33* | rs73181194-A | PIM3 | regulatory | 23.6% | CDTA | 1.08 | (1.05–1.11) | 1.50E-09 | CDTA | 0.99 | (0.90–1.10) | 0.91 |
| 16p11.2 | rs6565189-T | ITGAL | intronic | 29.1% | Pharyngeal** | 1.06 | (1.04–1.09) | 1.97E-09 | CDTA | 1.00 | (0.91–1.09) | 0.96 |
| 17q21.1* | rs3816470-A | IKZF3 | intronic | 43.2% | CISD** | 1.07 | (1.04–1.10) | 1.57E-08 | CRS** | 1.04 | (1.00–1.08) | 0.034 |
| 14q31.1* | rs1022434-A | NRXN3 | intronic | 7.1% | Sinonasal** | 0.89 | (0.85–0.93) | 3.47E-09 | NSD** | 0.97 | (0.90–1.05) | 0.47 |
| 5p13.2 | rs6897932-T | IL7R | intronic | 33.1% | IURD** | 0.96 | (0.94–0.98) | 3.26E-08 | NP** | 0.94 | (0.89–0.99) | 0.010 |
| 11q12.2* | rs174605-G | FADS2 | intronic | 25.4% | NP (MTAG) | 0.98 | (0.97–0.99) | 2.72E-08 | NP | 0.95 | (0.90–1.00) | 0.028 |

Loci lead variants are either replicated in UKB ($p < 0.05$) or multiple testing significantly ($p < 5e-9$) associated in FinnGen with no observed impact in UKB. "Consequence" is the most severe consequence annotated by the Variant Effect Predictor (VEP). Odds ratios (OR) were estimated using logistic regression (Methods). $P$ values were calculated using upper tail chi-square testing (one degree of freedom) from a t-statistic under a normal approximation. 95% CI were derived using normal approximation. *locus has no previous association with IURDs **Lead variants detected in cross-trait groups [Pharyngeal diseases (Pharyngeal), CISD, Sinonasal diseases (Sinonasal), IURD] are compared to UKB analysis of the specific IURD with smallest $p$ value in FinnGen.

OR and $p < 0.01$ in our analyses. In this way, our VAR GWAS replicated 18 of the previously reported[41,44–48] 34 allergic rhinitis loci with lead variants genotyped in FinnGen. Similarly, all 10 loci previously associated[39] with CRS and NP were replicated in our respective GWASs. This included the protective missense variant rs34210653-A in *ALOX15*, associated with NP [OR = 0.52 (0.38–0.71), $p = 1.62$E-05] and CRS [OR = 0.81 (0.67–0.97), $p = 0.016$]. Of the 35 tonsillectomy-associated loci, 26 had a corresponding variant genotyped in FinnGen, and of these 26 loci, our CDTA and PA analyses replicated 22. Finally, 2 of 2 loci associated with strep throat[40] also replicated in our CDTA and PA analyses.

## Characterization of loci

The nasal GWASs included five diagnostic groups: VAR, CRNP, CRS, NP, and NSD. Combined these identified 16 loci (Tables 2–3), of which four [2q12.1, 5q22.1, 6p21-22 (*HLA*), and 9p24.1] were MTS associated with VAR, CRS, and NP. These four loci have also been previously associated with asthma, allergic rhinitis, and eczema[44,45]. VAR also associated with the previously reported vitiligo locus[56] 9q33.3 near *NEK6* (lead variant rs3758213-T). CRS was associated with three non-*HLA* loci previously linked[39] with chronic rhinosinusitis without polyps (CRSwNP) and one with childhood ear infections[40]. NP was associated with thirteen loci, two of which have not been previously reported[39] (1q21.3 [$OR_{fg} = 1.16$ (1.11–1.23) ($OR_{uk} = 1.06$)] near *ARNT* and 2q22.3 [$OR_{meta} = 0.81$ (0.76–0.87)] near *ZEB2*. In addition, a missense variant in *GAS2L2* (rs3744374-A), with no previous associations, was protective of CRS [$OR_{fg} = 0.90$ (0.86–0.93)].

In the laryngotracheal area, we analyzed three diagnostic groups: CDTA, PA, and CLT. CDTA was associated with 15 non-*HLA* loci (Tables 2–3), of which eight have been previously linked with tonsillectomy[40], and one (1p36.23, near *SLC45A1*) with strep throat[40]. We also detected six CDTA loci not previously reported with tonsillar endpoints. These included two credible sets at the locus 2q13 with exonic variants in the long non-coding RNA *MIR4435-2HG*, previously shown to regulate myeloid cell proliferation in mouse models[57,58]. PA was associated with three loci linked with tonsillectomy[40], and two GWS associations not previously reported: 3q12.3 near *NFKBIZ* [$OR_{meta} = 1.13$ (1.08–1.17)] overlapping a previously reported psoriasis locus[59] and proximal to a COVID-19 susceptibility locus[18].

In the oral diseases, we observed one MTS locus for DPPT implicating *HORMAD2* with a credible set overlapping that of CDTA at the same locus (Supplementary Data 1). GWAS of DPPT subphenotypes repeated the 22q12.2 lead variant as GWS in pulpitis (K04.0, $n = 18,139$) and necrosis of pulp (K04.1, $n = 10,168$).

## Non-synonymous variants

To identify non-synonymous coding variants, we used SuSiE software[60] to fine-map credible sets of causal variants in the associated loci. Fine-mapping of IURD GWASs identified 42 credible sets with at least one GWS variant. We detected three loci with more than one such credible set. The fine-mapped credible sets included non-synonymous variants in nine protein-coding genes (Table 4). The *IL1RL1* and *ZPBP2* missense variants have been previously associated with Type 2 high childhood asthma[61] and adult-onset asthma[62], respectively. The *SLC22A4* and *FUT2* variants have been linked with Crohn's disease[63,64] with no previous IURD association. The *GSDMB* variant (rs2305479-T) was part of the same credible set as the asthma-linked *ZPBP2* missense variant rs11557467-T with high LD ($r = 95\%$) and lower posterior probability (1.5% for *GSDMB* vs 4.0% for *ZPBP2*).

Non-synonymous variants are also mapped to three known immune deficiency genes. The CDTA-associated 17p11.2 locus, previously also linked with tonsillectomy[40], identifies the non-synonymous variant rs72553883-T. This Finnish-enriched missense variant in the gene *TNFRSF13B* (encoding the protein TACI)[65,66] is linked to common variable immune deficiency (CVID) (variant MIM no

**Table 4 | Nine non-synonymous variants in protein-coding genes included in 95% credible sets with at least one GWS SNP**

| LOCUS | GWAS | RSID | OR | 95% CI | P value | EAF | FE | Gene | CONSEQ | PP |
|---|---|---|---|---|---|---|---|---|---|---|
| 2q12.1 | CRS | rs1041973-A | 0.90 | (0.87–0.94) | 6.21E-08 | 20.5% | 0.91 | *IL1RL1* | missense | 1.7% |
| 4q24a | IURD | rs2272676-T | 0.96 | (0.94–0.98) | 7.84E-09 | 34.9% | 1.10 | *NFKB1* | splice donor | 0.3% |
| 4q24b | CDTA | rs2454206-G | 0.94 | (0.92–0.96) | 5.59E-09 | 34.1% | 0.89 | *TET2* | missense | 0.4% |
| 5p13.2 | IURD | rs6897932-T | 0.96 | (0.94–0.98) | 3.26E-08 | 33.1% | 1.28 | *IL7R* | missense | 4.5% |
| 5q31.1b | NP | rs1050152-T | 0.86 | (0.81–0.91) | 3.73E-09 | 31.7% | 0.72 | *SLC22A4* | missense | 3.2% |
| 17p11.2 | CDTA | rs72553883-T | 1.43 | (1.33–1.53) | 2.39E-26 | 2.4% | **3.52** | *TNFRSF13B* | missense | 48.2% |
| 17q12 | CRS | rs3744374-A | 0.90 | (0.86–0.93) | 9.94E-10 | 23.0% | 0.95 | *GAS2L2* | missense | 99.9% |
| 17q21.1 | CISD | rs11557467-T | 0.94 | (0.91–0.96) | 1.60E-08 | 55.9% | 1.14 | *ZPBP2* | missense | 4.0% |
| 17q21.1 | CISD | rs2305479-T | 0.94 | (0.91–0.96) | 2.69E-08 | 54.8% | 1.14 | *GSDMB* | missense | 1.5% |

GWAS column denotes the genome-wide association study where the variant is identified. Odds ratios and p-values are with regard to the phenotype in GWAS column; the most specific phenotype is represented if the variant appears in several GWAS. Odds ratios (OR) were estimated using logistic regression (Methods). *P*-values were calculated using upper tail chi-square testing (one degree of freedom) from a t-statistic under a normal approximation. 95% CI were derived using normal approximation. *EAF* effect allele frequency, *FE* enrichment in FinnGen (Finnish-enriched variant is bolded), i.e., allele frequency compared with non-Finnish participants in gnomAD, *CONSEQ* most severe consequence annotated with (VEP), *PP* Posterior probability in fine-mapped credible set.

604907.0002) and primary antibody deficiency[67]. The missense variant rs2272676-T in CVID-linked *NFKB1*[68] decreases risk for CDTA. A missense variant in exon 6 of the severe combined immunodeficiency-linked[69] *IL7R* (phenotype MIM no. 608971) decreases risk for IURDs [OR = 0.96 (0.94–0.98)]. In total, IURD-associated non-synonymous variants in three genes—*NFKB1*, *IL7R*, and *TNFRSF13B*—are included in the IUIS list of Mendelian immune disorder genes[15].

### in silico analyses

Next, we used an in-house pipeline based on eCAVIAR[70] to evaluate the impact on gene expression (Methods). In brief, we colocalized fine-mapped credible sets of IURD GWAS summary statistics with similarly fine-mapped credible sets of eQTLs in GTEx v8[71] and the eQTL catalog[72] databases (Supplementary Data 7). Out of the 40 non-*HLA* IURD loci, eight paired with eQTLs in 42 tissues (excluding the CNS and gonads) with greater than 60% posterior agreement. Of interest is that three loci had at least 80% causal posterior agreement with credible sets of eQTLs in immunological cell types: the CDTA-associated (OR$_{fg}$ = 1.10) 2q13 locus decreased expression of *MIR4435-2HG* in lymphoblastoid cells and CD14+ CD16− classical monocytes; the CDTA- and PA-associated (OR$_{fg}$ = 1.16) 12p13.31 increased *LTBR* expression in macrophages, CD14+ CD16− classical monocytes, lymphoblastoid cells and T cells; and the NP-associated (OR$_{meta}$ = 1.15) locus 12q13.2 associated with increased *RAB5B* expression in CD4 + αβ-T cells and neutrophils. These links to the immune system are in line with identified shared IURD loci, and provide clues for an expected impact on gene expression in the relevant tissues themselves (tonsillar lymphoid tissue and upper respiratory epithelium).

We also tested gene enrichment with MAGMA[73] using summary statistics for all IURD phenotypes (Methods). We detected 96 gene-phenotype associations (Supplementary Data 8) for 74 genes, after correcting for multiple testing. MAGMA detected significant variant enrichment within 44 genes that were within the non-*HLA* GWS loci identified in the IURD GWAS of FinnGen (see above). Eleven genes were enriched in more than one IURD phenotype. Two genes, *WDR36* and *TSLP*, were associated with VAR, CRS, and NP through the shared locus on 5q22.1. Thirteen associated genes were not close to any of the GWS loci, including *IL2RB* (linked with NP and CRS), *ST5* and *ESR1* (both linked with CDTA), and a cluster of four genes at 20q13.33 associated with VAR.

We next tested for enrichment of genes in 4,761 curated gene sets and 5,917 GO terms. Enrichment of MAGMA-identified genes highlighted gene sets involved with immune function (Table 5), including major histocompatibility class II receptor activity, and regulation and production of interleukins 4 and 13. The tumor necrosis factor 2 pathway, which spans 16 recognized genes, includes 10 genes associated with CDTA. The recognized associations encompass variants in

genes *TRAF2*, *TRAF3*, *TANK*, *TNFRSF1B*, and *RIPK1*, all involved in producing the intracellular components of TNF receptor 2.

### Shared heritability with extended phenotypes

To evaluate shared impact on non-IURD phenotypes, we also here used an in-house pipeline based on eCAVIAR[70] to evaluate colocalization of IURD loci and 2,861 endpoints in the FinnGen PheWeb. We observed overlapping SNPs between credible sets of 19 non-*HLA* IURD loci, and a total of 319 credible sets from 95 FinnGen endpoints (Table 6). Causal posterior probability was >80% for colocalization between the NP locus 5q22.1 (*TSLP/WDR*) and asthma endpoints, and between the CDTA locus 12p13.31 (*LTBR*) and acute appendicitis. In addition, posterior probability was >20% between the CDTA locus 17p11.2 (*TNFRSF13B*) and non-suppurative otitis media, and the NP locus 9p24.1 (*IL33*) and asthma. Beyond these, causal posterior agreement was >50% for ten IURD loci and 56 non-IURD phenotypes. These 56 phenotypes include infectious and inflammatory disorders of the upper respiratory tract that were not included in our definition of IURD: acute sinusitis (9p24.1 near *IL33* and 17q21.1 near *IKZF3*) and acute respiratory infections (5q22.1 near *TSLP/WDR* and 9p24.1 near *IL33*). Asthma endpoints colocalized with seven non-*HLA* loci associated with NP, CRS, and VAR—similar to previous observations (CRSwNP)[39]. Association to type 1 diabetes and hypothyroidism was observed near *RAB5B* (12q13.2) and *HORMAD2* (22q12.2), and association to inflammatory bowel disease near *EMSY*, with overlapping credible sets near *SLC22A4* and *IKZF3*. The CDTA-associated locus near *ABO* colocalized with deep vein thrombosis, gastroduodenal ulcer, and type 2 diabetes. Broad colocalization was also observed for the VAR locus 11q13.5 (near *EMSY*), which colocalized with atopic dermatitis, conjunctivitis, and inflammatory bowel diseases in addition to asthma.

Genetic correlation analysis beyond IURD highlighted genome-wide links with susceptibility to infection, asthma, and allergic diseases. We analyzed genetic correlation using LD Score regression[53], estimating correlating impacts between IURD phenotypes and diseases associated in PheWAS analysis (Supplementary Figure 4). Sino-nasal diseases in particular formed a cluster of genetic correlation with asthma, allergic conjunctivitis, and atopic dermatitis (Supplementary Figure 4A). The recurring links to autoimmune diseases in several loci translated to genetic correlation with rheumatoid arthritis, mainly with the oral DPPT phenotype [$r_g$ = 58.6% (95% CI 27.7–89.5%); $p = 0.00020$] and CRS [$r_g$ = 57.6% (27.4–87.1%); $p = 0.00020$] (Supplementary Figure 4B). Other tested diseases, such as non-suppurative otitis media and sleep apnea, largely clustered separately despite significant correlations to specific IURD phenotypes (Supplementary Figure 4C). Finally, when comparing to PheWAS-linked inflammatory intestinal diseases, CRS and CDTA showed genetic correlation with diverticular disease and appendicitis (Supplementary Figure 4D).

**Table 5 | Gene sets enriched in MAGMA analyses**

| Phenotype | Gene set | Genes | β | $P_{adj}$ |
|---|---|---|---|---|
| CRS | MHC class II receptor activity | 9 | 1.6939 | 0.036904 |
| NP | MHC class II receptor activity | 9 | 2.5209 | 1.96E-07 |
| NP | MHC class II receptor complex | 14 | 1.7043 | 0.000367 |
| NP | Positive regulation of interleukin 13 production | 15 | 1.2187 | 0.000453 |
| NP | Regulation of interleukin-4 production | 28 | 0.90161 | 0.003912 |
| NP | Interleukin 13 production | 24 | 0.88367 | 0.004649 |
| NP | MHC protein complex | 22 | 1.0476 | 0.008563 |
| NP | Interleukin-4 production | 34 | 0.78389 | 0.011623 |
| CDTA | Cytokine-mediated signaling pathway | 748 | 0.16085 | 0.005677 |
| CDTA | TNFR2 pathway | 16 | 1.0288 | 0.00569 |
| CDTA | Reactome cytokine signaling in immune system | 835 | 0.14811 | 0.006434 |
| CDTA | Peptidyl serine autophosphorylation | 8 | 1.3374 | 0.027797 |
| PA | Negative regulation of morphogenesis of an epithelium | 16 | 1.12272 | 0.025657 |

Gene sets were identified based on genes identified as phenotype-associated in MAGMA analysis. β is the effect size in the MAGMA gene set enrichment analysis. The considered gene sets encompassed a set 4761 curated gene sets and 5917 Gene Ontology terms, as used in the FUMA pipeline. $P$ values were calculated using upper tail chi-square testing (one degree of freedom) from a t-statistic under a normal approximation. $P$ values have been adjusted for these sets (10,678 tests).

**Table 6 | IURD genomic loci credible sets shared with other FinnGen endpoints**

| Locus | Gene | IURD | Colocalized | Overlapped |
|---|---|---|---|---|
| 1q21.3 | ARNT | NP | - | Malignant neoplasm of skin (1) |
| 2q12.1 | IL18RAP | NP, VAR, CRS | Asthma (12); Allergic conjunctivitis (2) | Acute upper respiratory infections (2) |
| 2q13 | MIR4435-2HG | PA | - | Asthma (13); Umbilical hernia (2) |
| 4q24a | NFKB1 | CDTA, PA | - | Other diabetes (E13; 1) |
| 5q22.1 | WDR36 | NP, VAR, CRS | Asthma (15) | Polyp of the female genital tract (1) |
| 5q31.1b | SLC22A4 | NP | - | Asthma (9); Dermatitis (3); Breast cancer (2); IBD (2); Chalazion (1) |
| 9p24.1 | IL33 | VAR, CRS | Asthma (7); Acute sinusitis (1) | Asthma-related infections (4) |
| 9q33.3 | NEK6 | NP | - | Arthropathies (2) |
| 9q34.2 | ABO | CDTA | DVT of lower extremities and PE (3); Gastric ulcer (2); Type 2 diabetes (1) | Type 2 diabetes (3) |
| 10p14a | GATA3 | NP | - | Asthma (1) |
| 10p14b | GATA3 | NP, CRS | Asthma (5) | - |
| 11q13.5 | EMSY | VAR | Asthma (7); IBD (7); Allergic conjunctivitis (4); Atopic Dermatitis (4); Mucosal proctocolitis (1) | IBD (1) |
| 12p13.31 | LTBR | CDTA, PA | Appendicitis (2) | - |
| 12q13.2 | RAB5B | NP | Type 1 diabetes (4); Hypothyroidism (1) | Type 1 diabetes (4); Hypothyroidism (5) |
| 15q22.33 | SMAD3 | VAR | Asthma (5); Coronary revascularization (1); Haemmorrhoids (1); Allergy (1) | Asthma (8); Thyroid cancer (4) |
| 16p13.13 | CLEC16A | NP | Asthma (6) | Asthma (5); Type 1 diabetes (1) |
| 17p11.2 | TNFRSF13B | CDTA | Non-suppurative otitis media (1) | - |
| 17q21.1 | IKZF3 | CRS | Cervical cancer (2); Acute sinusitis (1) | Asthma (18); IBD (8); Mucosal proctocolitis (1) |
| 22q12.2 | HORMAD2 | CDTA | - | Type 1 diabetes (6); Hypothyroidism (4) |

Using an in-house colocalization pipeline (Methods), the phenotypes in the "IURD" column credible sets had significant causal posterior probability/agreement with non-IURD FinnGen phenotypes ("Colocalized"), or otherwise overlapping variants in credible sets with non-IURD FinnGen phenotypes ("Overlapped"). Non-IURD phenotypes in "Colocalized" and "Overlapped" columns are grouped according common traits, such that "Asthma" refers to endpoints such as "allergic asthma" or "childhood asthma", with the number of parallel endpoints included in a parenthesis. Broader categories (e.g., "Diseases of the respiratory system") are omitted. "Gene" reports the gene nearest to the lead variant, as reported in Tables 2 and 3. Full colocalization results are reported in Supplementary Data 9. NP Nasal polyposis, VAR vasomotor and allergic rhinitis, CRS chronic rhinosinusitis, CDTA chronic diseases of tonsils and adenoids, PA peritonsillar abscess, DVT deep vein thrombosis, PE pulmonary embolism, E13 ICD-10 code for "Other specified diabetes mellitus".

Since our investigation focused on host susceptibility to infection, we also compared our results to the summary statistics of the COVID-19 host genetics initiative[18], noting two shared loci at NFKBIZ and ABO. We investigated the genetic correlation of IURDs and their associated FinnGen endpoints, based largely on pre-pandemic diagnoses, with three COVID-19 endpoints. COVID-19 hospitalization (B2) in particular linked with CPs [$r_g$ = 57.2% (10.7–100.0); $p$ = 0.016], DPPT [$r_g$ = 41.8% (12.5–71.0); $p$ = 0.0051], all pneumonias [$r_g$ = 34.9% (3.2–66.7); $p$ = 0.031], CRNP [$r_g$ = 42.6% (11.7–73.4); $p$ = 0.0068] and hospital discharge record of unspecified acute upper respiratory infections [$r_g$ = 55.8% (16.3–95.3); $p$ = 0.0055] (Supplementary Figure 5). Pharyngeal or sinonasal diseases were not significantly associated.

## Discussion

To understand the genetic predisposition landscape of infectious and inflammatory upper respiratory diseases, we genome-wide analyzed these diseases both individually and in groups. In total, we identified 41 loci, of which twelve have not been previously reported to associate with any of the IURDs. Among the 41 loci, our fine-mapped credible sets identified nine coding variants. Cross-disease analyses combined

chronic diseases of the sinonasal, oral and pharyngeal regions. We showed that genetic structure distinguished sinonasal diseases and pharyngeal diseases, with a partly overlapping genetic background. Our study also includes the first GWASs of CDTAs, PA, and DPPT.

Our findings indicate overlapping genetic etiologies that extend beyond the previously reported genetic links between CRS and nasal polyps[39] to tonsillar endpoints as well. We observe both locus-specific and genome-wide correlations between sinonasal, oral, and pharyngeal inflammatory and infectious conditions. The associations implicate immunological pathways and links to immune-mediated diseases beyond the confines of the upper respiratory system.

Of the 40 reported non-HLA loci, 17 were uniquely observed in a single IURD phenotype GWAS. The remaining 23 loci had highly similar odds ratios in two or more phenotypes, even if the signal did not reach the genome-wide significance threshold in all diseases. This is in line with previous epidemiological and histopathological evidence that highlights links between inflammatory sinonasal diseases[5,6,26,44]. Similarly, pharyngeal diseases associate with eleven loci previously linked to tonsillectomy[40,41] as well as one locus associated with self-reported strep throat and childhood ear infections[40]. While there is a shared genetic contribution for three loci to all IURDs, the underlying genetic structure distinguishes between sinonasal diseases and pharyngeal diseases. This is supported by several lines of evidence, from genome-wide correlation to loci-specific log-odds-based hierarchical clusters and Bayesian meta-analysis. In addition to the known links among CISDs, we observe a shared heritability between the clinically distinct chronic (CDTA) and acute (PA) pharyngeal diseases, including sixteen loci with shared impact.

Overall, genetic correlation analysis illustrated the genetic landscape linking IURDs with asthma, reflecting the co-existence of IURDs and asthma; about half of AR, CRS, and NP patients have asthma[6]. We detected shared genetic risk for NP and CRS, which is in line with previous observations[39]. In subjects with NP, gene set enrichment associated pathways related to regulation and production of interleukins 4 and 13, which are hallmarks of Type 2 inflammation[74] and have shown to be associated with asthma[75] and to be functionally relevant in CRSwNP patients[76]. Ex vivo cultured nasal basal cells have been shown to retain intrinsic Type 2 high memory of IL-4/IL-13 exposure, which could be decoded via clinical blockade of the IL-4 receptor α-subunit in vivo[76]. We demonstrated a genetic landscape linking IURDs (such as NP, CRS, and AR) with asthma and allergic diseases[39,44–46]. We found that CISDs associate with the 17q21 locus (GSDMB/ZPBP2) as well as TSLP, IL33, and the gene encoding the IL33 receptor, IL1RL, which all have previously shown to be associated with asthma[77,78], and have also shown to be functionally relevant in asthma models[79–82].

Pharyngeal diseases implicate genes linked with immune deficiency. Non-synonymous variants were implicated in eight loci, highlighting three genes linked with immune deficiency and immune-mediated disorders. Interestingly, in two of these genes (NFKB1 and IL7R) with previously established risk variants for immune deficiency[15], we identify non-synonymous variants with decreased risk for IURDs.

Beyond the above-described trends, there were also associations with autoimmune disorders. Diseases such as rheumatoid arthritis correlated genome-wide with CRS and DPPT, highlighting the multitude of immunopathological mechanisms with manifestations in the upper respiratory tract. Links to immune-mediated disorders, such as asthma and inflammatory bowel diseases, were also observed in colocalization analysis. Among specific pathways, we implicate the tumor necrosis factor 2 pathway as a viable target for further study in the analysis of CDTAs. This furthers the findings of Tian et al.[40], who previously identified genetic links between tonsillectomy and the intestinal immune network for IgA production. We also replicated a shared locus near HORMAD2 between CDTA and type 1 diabetes, and extend IBD-associated[83] CDTA loci to SLC45A1 and PIM3. Beyond specific loci, we reported enrichment of CDTA-associated variants in 10 out of 16 genes involved with the TNFR2 pathway, and many

intracellular genes of the canonical NFκB pathway[84]. Notably, the TNFR2 pathway has been found to modulate allergic inflammation[85].

We observed shared impact with other infectious disorders. The immune deficiency was also evident in loci-specific impact on infectious disorders, specifically non-suppurative otitis media, and appendicitis. These two infections are also genetically correlated with CRS and CDTA. Two of the loci described herein—the ABO cluster that associates with CDTA and the PA locus closest to the gene NFKBIZ— have been implicated with COVID-19 severity[18,86]. Phenome-wide colocalization analysis of the ABO gene cluster shows wide-ranging phenotypic implications (Supplementary Data 9), in line with its well-described pleiotropic effects[87]. The fine-mapped set of CDTA-linked variants near ABO includes rs923383567-C [a.k.a. rs657152-C, linked with COVID-19 severity[86]] and rs879055593-C, the latter of which was linked with interleukin-4 driven pathogenesis in a recent multitrait analysis[88]. The NFKBIZ locus has two SNPs with near-equal posterior inclusion probability in fine-mapping: rs1456200-A and rs1456202-G (Supplementary Figure 6). The cryptic LD structure suggests that further work is needed to fine-map this region. While pharyngeal diseases showed little general genetic correlation with COVID-19 in this analysis (Supplementary Figure 5), it is interesting to note the genetic correlation with oral infections and CRNP, although only a few loci could be identified in these disorders. The implications of these results require further study and replication.

This study has some limitations. Firstly, the FinnGen study cohort is collected based on legacy samples variably representing certain aspects of the population, with new participants being recruited mainly in the University Hospital health care setting. For these reasons, the study cohort is not a true population sample and thus comorbidity analyses should not be interpreted from an epidemiological angle. Second, the IURD phenotypes are diagnosed by specialists, often in hospital settings, and thus likely quite accurate but are therefore subject to ascertainment bias (collider bias) with other disorders—a feature of study design that can inflate correlation estimates with other diseases. Thirdly, the ICD-10-based disease endpoints used here differ somewhat from current clinical practice (e.g., non-allergic rhinitis and AR, CRSsNP, and CRSwNP). Also, phenotypic coding definitions differed somewhat between FinnGen vs. UKB studies. However, as the genetic association to disease biology does not necessarily follow clinical manifestations, and there is notable previous success using this approach[39], we, therefore, find these categories appropriate to highlight the genetic similarities and differences. Finally, while the VAR and NP analyses in the UKB were well-powered for replication, the effective sample sizes for other IURDs were not sufficient for reliable replication analysis of many of the lead variants. An inherent feature of genetic association analysis in population isolates is that loci identified with population-specific enriched variants are hard or impossible to be analyzed adequately in more mixed populations. A non-replication does not necessarily mean a false positive.

Using lifelong national register data, we identified 41 loci associated with different upper respiratory diseases. These loci identified genes involved in immunological (such as Type 2) mechanisms and immune-mediated diseases. We observed both shared and distinct genetic contributions among different chronic inflammatory upper respiratory diseases, between IURDs and other systemic immune-mediated disorders, and between IURDs and two oral inflammatory diseases, providing genetic insight into earlier clinical and epidemiological observations.

## Methods
### Study design
The FinnGen study is an on-going nationwide collection of Finnish genetic samples, combining genome information with digital health care and registry data. Participants include legacy samples from previous studies recruited for on-going research, maintained by the

Biobank of the Finnish Institute for Health and Welfare (THL), and recent biobank samples recruited at university hospitals across Finland. In the present study, we included samples from 271,341 participants released in August 2020. Registry data used here included disease diagnoses and performed operations from the Care Register for Health Care (THL), the Primary Health Care Register (THL), the Causes of Death Register (Statistics Finland), and the Drug Reimbursement Register (KELA, the Social Insurance Institution of Finland). The co-occurrence of the IURD diagnoses is summarized in Supplementary Data 10.

Participants in FinnGen provided informed consent for biobank research, based on the Finnish Biobank Act. Alternatively, separate research cohorts, collected prior to the Finnish Biobank Act came into effect (in September 2013) and the start of FinnGen (August 2017), were collected based on study-specific consents and later transferred to the Finnish biobanks after approval by Fimea, the National Supervisory Authority for Welfare and Health. Recruitment protocols followed the biobank protocols approved by Fimea. The Coordinating Ethics Committee of the Hospital District of Helsinki and Uusimaa (HUS) approved the FinnGen study protocol Nr HUS/990/2017. The FinnGen study approval permits are listed in Supplementary Table 4, and biobank access decisions are listed in Supplementary Table 5.

### Genotyping and sample quality control
Samples were genotyped using called for a total of 271,341 individuals. In total, 12 different type of DNA chips were used to analyze participants in 78 batches. In genotyping, we removed SNPs with high missingness (>2%), minor allele count <3, and Hardy–Weinberg equilibrium ($p_{HWE} < 1e-6$). We removed samples with non-Finnish heritage in PC analysis, duplicated/twins, incomplete phenotypic information, or mismatch between reported and imputed genetic sex. The final post-QC sample count was 260,405 (147,061 females and 113,344 males). Genotypes were imputed based on a Finnish reference panel detailed elsewhere[89], using deep whole-genome sequencing data from 3775 Finns in the SISu[90] reference panel.

### Genome-wide association analyses
We used the SAIGE software[49] for running mixed model logistic regression genome-wide on 16,355,289 variants. We used age, sex, the first 10 PCs, and genotyping batch as covariates. We analyzed the genome-wide association of cases of eight IURDs (Table 1; total $n = 61,197$) against 199,208 controls with no IURDs in the FinnGen dataset. The IURD phenotypes did not include J38 ("Diseases of vocal cords and larynx, not elsewhere classified") and J39 ("Other diseases of upper respiratory tract"), which were included in the larger IURD category but were not separately analyzed. Two oral phenotypes, DPPT and CP, were separately analyzed due to epidemiological overlap with pharyngeal and sinonasal IURDs. The study-wide level of significance (MTS) was set at 5e-9 to correct for the simultaneous analysis of ten different diseases. Non-MTS GWS loci (5e-9 < $p$ < 5e-8) were considered significant only if they replicated in UKB or meta-analysis.

### Cross-trait analysis
To investigate the shared heritability across multiple IURDs, we performed cross-trait GWASs of the three disease clusters identified through genetic correlation (Fig. 2). These disease groups were sinonasal diseases ($n = 25,235$ cases vs 199,208 controls), pharyngeal diseases ($n = 33,157$ cases vs 199,208 controls), and CISD ($n = 19,901$ cases vs 199,208 controls). In addition, we performed a GWAS of cases with any IURD ($n = 61,197$ vs 199,208 controls). The cross-trait analyses were run using the same SAIGE pipeline as the main analyses, with the same covariates. We performed MultiTrait Analysis of GWAS (MTAG)[54] using GWAS summary statistics from all IURD phenotype GWASs jointly. Only variants with MAF >1% were considered ($n = 6,868,381$). As the approach has an elevated type II error rate, loci identified by MTAG were only considered meaningful if replicated in the UKB analysis.

### Bayesian analysis of shared variant effects
In order to estimate the shared and distinct phenotypic impacts of specific loci in our GWAS results, we used a Bayesian framework, where we assessed for a shared effect between phenotypes. This framework adjusted for overlapping controls using a previously reported variance-based adjustment[55]. The framework considered three types of impact: none (the "null model"), unique ("one phenotype only"), and shared. Shared impact combined a hypothesized "fixed" model, where the variant has the same effect size for all phenotypes, and a "correlated" model, where the variant has similar, but not necessarily the same, effects for all phenotypes. The posterior probability of the "fixed" and "correlated" models were added together and called the "shared" model when comparing with the null model and the unique effects models. The prior probability of "fixed" and "correlated" models were half of that of "null" and "one phenotype only" models so that each of the compared models (null, shared, and one phenotype only models) had the same prior probability. We interpreted a model with at least 70% posterior probability as "most probable" model.

Shared impact between IURDs was analyzed in four tiers, corresponding to the four phenotype groups (IURDs, and sinonasal, pharyngeal, and allergic sinonasal diseases). The first tier analyzed heterogeneity of GWS SNPs identified in IURD GWAS based on their impact on CRNP, sinonasal diseases, pharyngeal diseases, and CLT. The second tier analyzed heterogeneity of GWS SNPs identified in the sinonasal disease GWAS on their impact to NSD and CISD. The third tier analyzed the heterogeneity of GWS SNPs of the tonsil disease GWAS on CDTA and PA. The fourth and final tier analyzed the heterogeneity of GWS SNPs of the CISD GWAS on VAR, CRS, and NP.

### Comparison with UK Biobank
For replication, we analyzed the association of lead variants of all 59 GWS non-HLA loci in the UK Biobank (Supplementary Data 1). We mapped the IURD and oral endpoints in FinnGen to corresponding UKB read codes (Supplementary Data 11) using ICD-to-Phecode mapping (in addition to manual curation of these codes based on description) using hospital data, cause of death registry, and for a subset ($n = 230,000$) also GP data. UKB variants were aligned to variants in FinnGen. In case of no exact match between SNPs (ref and alt differ between studies), matching was tried by flipping strand and/or switching ref->alt and alt->ref for the UKB variant. Variants were tested against the constructed endpoints in the UKB European population using using logistic regression. Covariates were reported gender, baseline age, and PCs 1–10. We used as controls all UKB participants with no identified IURD or oral endpoint ($n = 298,846$). To estimate heterogeneity of effect between the cohorts, a test statistic was calculated with the formula

$$z = \frac{(\beta_1 - \beta_2)^2}{SE_1^2 + SE_2^2} \tag{1}$$

where $\beta_i$ is the effect size of study $i$, and $SE_i$ is the standard error of the effect estimate in study $i$. The test statistic $z$ was assumed to follow a $\chi^2$ distribution with one degree of freedom. Variants with a $p_z < 0.05$ were considered heterogeneous. Only co-directional variants were meta-analyzed. We used inverse-variance-weighted meta-analysis under a fixed-effect assumption. Variants were considered significant if they had a GWS impact after meta-analysis with UKB, had a GWS association in FinnGen, and replicated a co-directional association ($p < 0.05$) in UKB, or MTS impact in FinnGen alone.

### Characterization of loci
After the initial detection of GWS ($p < 5e-8$) associated SNPs, we chose lead SNPs based on lowest $p$ value, and GWS SNPs in the same locus were grouped based on genomic distance <2 Mb, $r^2 > 0.1$ with lead SNP. We used SuSiE[60] for detection of credible sets of causal variants, with a

Finnish-based reference panel [Sequencing Initiative Suomi[90]] for LD structure and imputation. Only credible sets with at least one GWS ($p < 5e\text{-}8$) SNP were considered.

## Colocalization analyses

We used an in-house pipeline based on eCAVIAR[70] to evaluate colocalization with GWAS summary statistics. The pipeline uses SuSiE-fine-mapped CSs as inputs, and calculates a causal posterior probability (CLPP) that the same locus is causal in both studies. CLPP is defined as the sum of the products of SuSiE-fine-mapped posterior probabilities (PIP; $x$ for phenotype 1, $y$ for phenotype 2) for each variant $i$ shared in credible sets of both phenotypes, such that for credible set $k$:

$$\text{CLPP}_k = \sum_{i \in \text{CS}} x_i{}^* y_i \qquad (2)$$

CLPP was considered significant if it was higher than 20%. Another colocalization metric, causal posterior agreement (CLPA), was devised as a metric independent of CS size. CLPA represents the agreement between the fine-mapping results in both studies, and is defined as the sum of minimum PIP of shared variants between CSs from phenotype 1 and 2. CLPA was considered significant if higher than 50%. For gene expression, the GWAS summary statistics were derived from GTEx v8[71] and the eQTL catalog[72]. Phenotype summary statistics were derived from the FinnGen PheWeb.

## LD score regression

We estimated both the SNP-based observed scale heritability and genetic correlation ($r_g$) by performing LD Score Regression using the LDSC toolset[53]. This method works by using an "LD score" to estimate the amount of linkage disequilibrium (LD) each SNP has with the rest of the genome under the polygenic model, and regresses the $\chi^2$-statistic from a GWAS on the LD score, which also allows the estimation confounding bias. For our analysis, we used a previously calculated LD structure distributed by the ldsc.py software package. The distributed LD structure is based on the 1KG Phase 3 European dataset, and was merged to LD scores with the HapMap v3 variants[91]. The GWAS summary statistics were merged with 1,217,311 SNPs for which the LD scores were precalculated. 1073 SNPs were removed due to being strand-ambiguous, 1328 SNPS were removed due to duplicated rs-numbers, and 594 SNPs due to differing FinnGen and HapMap annotation. The remaining 1,190,282 SNPs were used in all FinnGen genetic correlation calculations.

LD score regression is developed for use in logistic and linear regression GWAS, while a GWAS using the SAIGE mixed model is not applicable for heritability estimates[49]. Therefore, the observed scale SNP-based heritability estimates were calculated using summary statistics from separate GWASs, in turn, run using independent subsets for all phenotypes and using standard logistic regression. For the heritability analyses, a total of 54,784 SNPs were removed from the initial 1,217,311 SNPs used for reference, and the observed scaled heritability estimate was calculated from the remaining 1,162,527 SNPs. In the logistic regression GWASs, we again used age, sex, PCs 1–10, and genotyping cohort as covariates.

We analyzed genetic correlation using LD Score regression to recognize shared heritability among IURD phenotypes. We grouped together phenotypes based on previously used thresholds, starting at $r_g > 75\%$. This grouped six of the eight IURD phenotypes into two groups: one group formed by VAR, CRS, NP, and NSD; another group being formed by CDTA and PA. Raising the threshold even further, to $r_g > 75\%$, distinguished a third group consisting of VAR, CRS, and NP. These groups were labeled "sinonasal diseases", "pharyngeal diseases" and "chronic inflammatory sinonasal diseases", respectively. We additionally analyzed genetic correlation to phenotypes detected in the phenome-wide analysis. Summary statistics for endpoints were

from FinnGen release 6, with the exception of inflammatory bowel disease where a previously published analysis[92] was used.

## Multi-marker Analysis of GenoMic Annotation (MAGMA)

We investigated gene- and gene-set enrichment separately using MAGMA[73]. Briefly, the pipeline calculates the mean $\chi^2$ statistic from IURD GWAS summary statistics per gene, and thus obtains a $p$ value for the gene. MAGMA was analyzed using the FUMA pipeline that tests association for 19,535 curated genes; thus, the adjusted $p$ value threshold was set to $p < 0.05/19,535 = 2.56e\text{-}6$. Genes with $p$ value below this threshold were considered to associate with the relevant phenotype. We again employed the UK Biobank release 2 reference panel, with 1000 randomly selected individuals for reference to reduce runtime. Gene analysis was performed with default FUMA parameters, only considering SNPs that overlap genes. The *HLA* region was not omitted from MAGMA runs. We next tested for enrichment of genes involved in 4761 curated gene sets and 5917 GO terms included in the FUMA pipeline. Here the level of statistical significance was set with Bonferroni correction at $p < 0.05/(4761 + 5917) = 4.68e\text{-}6$.

## Data availability

The summary statistics data generated in this study have been deposited in the FinnGen database (https://www.finngen.fi/en/access_results and http://r6.finngen.fi/). Individual-level genotypes and register data from FinnGen participants can be accessed by approved researchers via the Fingenious portal (https://site.fingenious.fi/en/) hosted by the Finnish Biobank Cooperative FinBB (https://finbb.fi/en/). Data release to FinBB is timed to the biannual public release of FinnGen summary results, which occurs twelve months after FinnGen consortium members can start working with the data. Freely available summary statistics data was obtained from the GWAS catalog (https://www.ebi.ac.uk/gwas/), the GTEx database (https://www.gtexportal.org/) and the eQTL catalog (https://www.ebi.ac.uk/eqtl).

## Code availability

Software used in this analysis is publicly available software distributed by the respective websites (SAIGE v0.39.1: https://www.leelabsg.org/software; SuSiE: https://github.com/stephenslab/susieR; LDSC v1.0.1: https://github.com/bulik/ldsc/) with developer pages on github. The Bayesian analysis framework is publicly detailed in the cited work, in addition to the website (https://github.com/trochet/metabf). Please see https://finngen.gitbook.io/documentation/ for a detailed description of data production and analysis including code used to run analyses. Please see https://github.com/FINNGEN/ for further code repositories used to run analyses in FinnGen. R code to reproduce figures is available upon request.

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

## Acknowledgements

The FinnGen project is funded by two grants from Business Finland (HUS 4685/31/2016 and UH 4386/31/2016) and the following industry part-ners: AbbVie Inc., AstraZeneca UK Ltd, Biogen MA Inc., Bristol Myers Squibb (and Celgene Corporation & Celgene International II Sàrl),

Genentech Inc., Merck Sharp & Dohme Corp, Pfizer Inc., GlaxoSmithKline Intellectual Property Development Ltd., Sanofi US Services Inc., Maze Therapeutics Inc., Janssen Biotech Inc, and Novartis AG. Following biobanks are acknowledged for delivering biobank samples to FinnGen: Auria Biobank (www.auria.fi/biopankki), THL Biobank (www.thl.fi/biobank), Helsinki Biobank (www.helsinginbiopankki.fi), Biobank Borealis of Northern Finland (https://www.ppshp.fi/Tutkimus-ja-opetus/Biopankki/Pages/Biobank-Borealis-briefly-in-English.aspx), Finnish Clinical Biobank Tampere (www.tays.fi/en-US/Research_and_development/Finnish_Clinical_Biobank_Tampere), Biobank of Eastern Finland (www.ita-suomenbiopankki.fi/en), Central Finland Biobank (www.ksshp.fi/fi-FI/Potilaalle/Biopankki), Finnish Red Cross Blood Service Biobank (www.veripalvelu.fi/verenluovutus/biopankkitoiminta) and Terveystalo Biobank (www.terveystalo.com/fi/Yritystietoa/Terveystalo-Biopankki/Biopankki/). All Finnish Biobanks are members of BBMRI.fi infrastructure (www.bbmri.fi). Finnish Biobank Cooperative -FINBB (https://finbb.fi/) is the coordinator of BBMRI-ERIC operations in Finland. The Finnish biobank data can be accessed through the Fingenious® services (https://site.fingenious.fi/en/) managed by FINBB. The UK Biobank replication research was conducted using the UK Biobank Resource under Application Number 22627. We thank the GWAS catalog (https://www.ebi.ac.uk/gwas/), the GTEx database (https://www.gtexportal.org/), and Tian and colleagues, for sharing expression data and summary statistics of GWAS studies. This study was funded by the Sigrid Juselius Foundation, the Horizon 2020 Research and Innovation Program [grant number 667301 (COSYN) to A.P.], the National Institute of Health (Grant no. 1U01MH105666-01), the Finnish Medical Foundation (Grant no. 3264), the Centre of Excellence Complex Disease Genetics (CoECDG, University of Helsinki, Academy of Finland Grant no. 312074 for A.P., Grant no. 312062 for S.R., Grant no. 312073 for J.K. and Grant no. 312075 for M.D.), and the Doctoral School for Population Health (University of Helsinki). A.S.H was supported by the Academy of Finland, Grant no. 321356, and S.R. was also supported by the Finnish Foundation for Cardiovascular Research and the University of Helsinki HiLIFE Fellow and Grand Challenge grants. The funders had no role in the study design, data collection, analysis, decision to publish, or preparation of the manuscript.

## Author contributions

Each author's contribution(s) to the paper is listed according to the CRediT model. Conceptualization: E.C.S., J.T.R., T.K., H.H., M.P., S.R.i., M.D., T.P., A.M., A.P. Methodology: M.K., M.P., H.H., J.M., J.K. Validation: J.M., M.W., N.M. Formal analysis: E.C.S., J.K., J.M., M.W., M.T., N.M., S.R.u. Investigation: J.K., FINNGEN Resources: FINNGEN Data Curation: J.K., T.K, A.S.H., FINNGEN Visualization: E.C.S. Funding acquisition: A.P., A.M., T.P., M.D., S.Ri., M.K., M.P., A.S.H. Project administration: A.P. Supervision: A.P., A.M., T.P., M.D., S.Ri., M.K., M.P., A.S.H. Writing—original draft: ECS Writing—review & editing: J.K., J.T.R., A.S.H., H.H., M.P., S.R.i., M.D., T.P., A.M., A.P., S.T.S.

## Competing interests

S.T.S. reports consultancies for AstraZeneca, ERT, Novartis, Sanofi Pharma, and Roche Products and a grant of GSK, outside the submitted work. All other authors declare competing interests.

## Additional information

[1]Institute for Molecular Medicine Finland (FIMM), HiLIFE, University of Helsinki, Helsinki, Finland. [2]Department of Otorhinolaryngology—Head and Neck Surgery, University of Helsinki and Helsinki University Hospital, Helsinki, Finland. [3]Stanley Center for Psychiatric Research, Broad Institute of MIT and Harvard, Cambridge, MA, USA. [4]Analytic and Translational Genetics Unit, Massachusetts General Hospital, Boston, MA, USA. [5]Cardiovascular Disease Initiative, Broad Institute of MIT and Harvard, Cambridge, MA, USA. [6]Finnish Institute for Health and Welfare, Helsinki, Finland. [7]Broad Institute of MIT and Harvard, Cambridge, MA, USA. [8]Skin and Allergy Hospital, Helsinki University Hospital and University of Helsinki, Helsinki, Finland. [9]Department of Mathematics and Statistics, University of Helsinki, Helsinki, Finland. [10]Department of Public Health, Faculty of Medicine, University of Helsinki, Helsinki, Finland. [11]Orthodontics, Department of Oral and Maxillofacial Diseases, Clinicum, Faculty of Medicine, University of Helsinki, Helsinki, Finland. [12]Oral and Maxillofacial Diseases, Helsinki University Hospital, Helsinki, Finland. [13]Analytic and Translational Genetics Unit, Department of Medicine, Department of Neurology and Department of Psychiatry, Massachusetts General Hospital, Boston, MA, USA. ✉e-mail: aarno.palotie@helsinki.fi

# FINNGEN

Elmo C. Saarentaus [1,2], Juha Karjalainen[1,3,4], Joel T. Rämö [1,5], Tuomo Kiiskinen [1,6], Aki S. Havulinna[1,6], Juha Mehtonen [1], Heidi Hautakangas [1], Sanni Ruotsalainen [1], Nina Mars[1,7], Sanna Toppila-Salmi[8], Matti Pirinen [1,9,10], Mitja Kurki[1,3,4], Samuli Ripatti [1,7,10], Mark Daly[1,4,7], Tuula Palotie[11,12], Antti Mäkitie [2] & Aarno Palotie [1,3,13] ✉

A full list of members and their affiliations appears in the Supplementary Information.

