## [Transparent Peer Review File · Nature Communications]

Inflammatory and infectious upper respiratory diseases associate with 41 genomic loci that link to type 2 inflammationREVIEWER COMMENTS

Reviewer #1 (Remarks to the Author: Overall significance):

Saarentaus and colleagues performed a genome wide association of combined set of infectious upper respiratory diseases (IURDs) in Finland. They claimed associations of variants with IURDs at 59 loci. Whereas it is clear that this represents a large amount of work, the current presentations has major limitations. First it is somewhat unclear what is really claimed, the combined set or the subsets of diseases. Second through the manuscript it is unclear what is novel and what is new; the reader has to arrive to the discussion to see that clearly articulated. I would have found easier to take specifically homogenic unit of disease(i.e.. the subset in the current), perform an international meta -analysis with foreign public set such as UKB (UK biobank) and/or BBJ (biobank japan). Then have as robust association as possible, i.e. choose more stringent cutoff than $5E-8$, a cutoff that have been proposed a long time ago under different settings (i.e., population, number of markers).

Reviewer #1 (Remarks to the Author: Impact):

I understand the work of the authors which is somewhat methodological when grouping all the IURDs phenotypes, but I feel that this demonstration somewhat limit the conclusions and the robustness of the results. If the authors succeed to have a clearer presentation including replication and combined analysis with abroad. A large number of the results presented here might be false positive, since 1)the cutoff used $5E-8$ had been set under different circumstances, population and number of markers 2) Many of the claimed association are less than an order of magnitude from the lenient threshold 3) most of them 24 out of 41 do not replicate nominally in their foreign group. Additionally, one would expect that the authors attempt replication of all the reported variants in the literature for the phenotypes they study, in order to asses how comparable their groups are with the one used for previously reported associations. For example, a variant in ALOX15 has been reported to confer a strong and significant protection against NP and CRS. It is somewhat unexpected not to see it mentioned in this manuscript.

Reviewer #1 (Remarks to the Author: Strength of the claims):

One of the main issue with this manuscript is the phenotype. I understand the angle the authors have taken: starting with a combination of phenotypes in Finland. But I truly am of the opinion that it makes it very unclear what is really claimed.

I would strongly suggest to start with specific phenotypes, performing meta-analysis with other population than Finland and with more stringent significance cutoffs to then arrive to a set of robust associations that they can then assess in the other subsets. In the UK biobank some of these diagnosis have to be derived from the GP sets in addition to hospital diagnosis.

Also the authors might realize that a lot of these diagnosis are differential diagnosis for each others, and that a large number of the patients probably overlap between subsets. Personally, I recommend to focus on the strong signal, choose more significant cutoffs . And present the results that are specific on the subsets, once combined with public datasets.

Reviewer #1 (Remarks to the Author: Reproducibility):

As mentioned in different places above, the cutoff for statistical significance of $5E-8$ has been proposed many years back at the time where sample size and number of markers tested were different. From the attempt of replication currently displayed it is possible that a large fraction of the associations are not true positive.

If they were, the ones very close to the cutoff are likely suffering from winner's curse, i.e. an inflation of effect.

Using meta-analysis with foreign data should be performed when available.

Again, I would rather have claims specific to the subsets that are robust presented first, with combined significance with the foreign groups.

I would present replications of all previous reported markers for these phenotypes.

Reviewer #2 (Remarks to the Author: Overall significance):

The manuscript by Saarentaus and colleagues describes a GWAS of inflammatory and infectious upper respiratory disease (IURD) in ~260k FinnGen study participants. They identify 59 genome-wide significant loci associated with this heterogeneous disease definition and/or component diseases of the umbrella term. I have a few comments for consideration by the authors:

1. Identifying loci by lumping together different diseases into a single 'mega case' group is a reasonable approach but I wonder if there are better alternatives. Have the authors explored using Genomic SEM (or similar approaches) to identify genetic factors that are shared across the individual disease GWAS?

2. The authors have effectively run 12 GWASs but corrected for only one ($P < 5 \times 10^{-8}$) from what I can see. They may want to justify this approach.

3. "Lead variants of the 58 non-HLA GWS loci were associated with 2,861 endpoints in FinnGen" – I suspect this number would be substantially lower if co-localization analyses were run. Presumably it's possible that many of these associations are in "LD shadows" of other signals?

4. "We then compared SNPs with expression data from GTEx v8 and DICE, which showed change in the expression of a total of 264 genes in all 58 non-HLA loci in total" – this statement really highlights the limitations of using tools like FUMA to assign causal genes to GWAS hits. The mapping of non-synonymous variants appeared methodologically robust, and I'd suggest the authors apply a similarly conservative approach to eQTL mapping. At a minimum the authors need to ensure they use eQTL integration approaches that reduce spurious associations due to more coincidental LD overlaps.

Reviewer #2 (Remarks to the Author: Impact):

Personally I feel this manuscript lacks the novel biological or epidemiological insights that warrant publication in journal such as Nature Genetics. A more incremental but rigorous study of this type may be better suited to Communications Biology or Nature Communications.

Reviewer #2 (Remarks to the Author: Reproducibility):

From what I can see it is not possible for anyone outside of FinnGen to access individual level data and reproduce these results. This is a shame given the precedent set by studies such as UK Biobank.

Reviewer #3 (Remarks to the Author: Overall significance):

The authors performed genome-wide association studies of a series of inflammatory and infectious upper respiratory diseases and identified 59 loci associated with one or more of these diseases, of which 23 are novel. They identified a high degree of sharing between many IURD subtypes, as well as with other related phenotypes such as asthma and allergies. The analyses are appropriate and well-performed and the paper is well-written. I have some questions and concerns that I hope the authors will be able to address:

Major points

1. On page 3, several previous genetic studies of IURDs are mentioned, but the authors do not seem to place their new results in the context of these studies. For instance, when compared to Kristjansson et al 2019, the current manuscript has twice the number of CRS cases and a similar number of NP cases, yet did not report associations at the loss-of-function variant in ALOX15. I would be curious to know whether these and other variants reported in the literature also showed associations in FinnGen, and if not, whether aspects of study design (e.g. case definition) or differences in LD patterns can lead to certain loci not being detected.

2. With regards to the replication work in UKB:

a) Page 7: "We found that the effect size differed ($p < 0.05$) only for the CRS association to the locus 5q22.1 (near WDR36) and the NP association to the locus 1q211.3 (near ARNT)."

In Table S6, it looks like the GAS2L2/TAF15 association with CRS was also not replicated and has a significant effect size difference in UKB.

b) In general, how should we interpret loci that failed replication, despite having over 95% power to do so at $\alpha=0.05$? Could it be that some of these diseases that occurred earlier in life are not well-captured in the UKB's cohort of middle-aged participants? Is there any evidence of different LD patterns between Finnish and UK population at some of the non-replicating loci?

c) Similarly, in UKB, the CDTA phenotype had the smallest effective sample size, while in FinnGen, this phenotype had the largest. Is this a consequence of diseases that occur earlier in life not being as well captured in UKB? If that is the case, how should we interpret the replication results in UKB more generally?

3. In the PheWAS analysis of lead SNPs, Sup table 7 shows 462 significant variant-phenotype pairs. How many of these pairs are actually being most likely driven by the same causal variant versus being driven by LD with another causal variant?

4. Several variants in Table 4 appear to have allele frequencies that are inconsistent with other tables.

For instance, rs189411872 has frequency 1.3% in Table S4, but 66% in Table S3.

5. In Table 3, please include the posterior probability of causality for each variant to help quantify the likelihood that the variant is causal. It also appears that rs11557467 and rs2305479 are two missense variants in different genes but belong to the same credible set? Please clarify.

Minor points

6. Page 7. "Lead variants of the 58 non-HLA GWS loci were associated with 2,861 endpoints in FinnGen". The phrasing here is ambiguous. On first reading I thought this meant that the 58 loci showed significant associations with 2861 other phenotypes, but I think this number refers to the total number of phenotypes tested?

7. Table S4. MAF column should be renamed EAF since many frequencies are > 0.5

8. Table S7. Are the betas with respect to the same ref/effect alleles given in Tables S1-S3? Please clarify

9. "SNP" is used throughout the paper even though many of the lead variants at significant loci are indels.

Reviewer #1 comments:

We thank Reviewer #1 for the valuable feedback. We find that approaching the topic from specific diagnoses and recompiling endpoints in the UK Biobank has improved the readability of our work. We think that the proposed replication of previous associations has improved both the robustness and reliability of our results, in addition to providing the reader with a clearer framework.

Saarentaus and colleagues performed a genome wide association of combined set of infectious upper respiratory diseases (IURDs) in Finland. They claimed associations of variants with IURDs at 59 loci. Whereas it is clear that this represents a large amount of work, the current presentation has major limitations. First it is somewhat unclear what is really claimed, the combined set or the subsets of diseases. Second through the manuscript it is unclear what is novel and what is new; the reader has to arrive to the discussion to see that clearly articulated. I would have found easier to take specifically homogenic unit of disease (i.e., the subset in the current), perform an international meta-analysis with foreign public set such as UKB (UK biobank) and/or BBJ (biobank japan). Then have as robust association as possible, i.e. choose more stringent cutoff than $5E-8$, a cutoff that have been proposed a long time ago under different settings (i.e., population, number of markers).

- We thank Reviewer #1 for this feedback. In order to more appropriately present our results, we have added major revisions to restructure the Results section of the manuscript. The new version first presents the specific phenotype analyses (nasal polyps, peritonsillar abscess), then investigates the shared heritability through genetic correlation and cross-trait analyses. We then present the Bayesian analysis pipeline to disentangle shared and distinct loci. We have also designed and performed our endpoints in the UKBB from read codes (incl GP data). In this way, we now report only loci that are either genome-wide significant in meta-analysis, multiple testing significant ($5e-9$ for ten independent phenotypes), or genome-wide significant with nominal replication in UKB. After a final set of variants this, previous literature is compared with, before characterization of loci and *in silico* follow-up analyses are presented.

The results section in particular now appears as follows (lines 93–105, p. 5):

“We performed genome-wide association of all IURD cases (n = 61,197, ranging from 2,623 to 29,135 per phenotype) in FinnGen (**Table 1**). We genotyped and imputed 16,355,289 single-nucleotide genetic variants in 260,405 Finnish individuals of all ages. We used a logistic mixed model with the SAIGE software⁴⁹ (see Methods) to detect genome-wide association between 61,197 cases of **different** IURD diagnoses (Table 1, Supplementary Figure 1) **using the same** 199,208 controls **for all IURDs**, and set as covariates age, genetic sex, principal components (PCs) 1–10, and genotyping batch. **[Moved:]** In addition to the main phenotypes linked to the upper respiratory tract, we also analyzed genome-wide association to two oral inflammatory diseases that have been associated⁵⁰⁻⁵² with IURDs: diseases of pulp and periapical tissues (DPPT; ICD-10 K04, **48,687 cases vs 211,718 controls**) and chronic periodontitis (CP; ICD-10 K05.30-.31, **14,631 cases vs 245,774 controls**). **We set the level of multiple testing significance (MTS) at $p < 5e-09$ for ten independent phenotypes. The eight different IURD GWASs detected 907 MTS variant associations in 25 independent loci in total (Supplementary Table 1).**”

We then present the shared heritability among IURDs. Figure 1 now features a heatmap of lead variant effect sizes for the 25 MTS loci (lines 107–113, p. 5–6):

“**Thirteen of the loci showed similar impact among IURDs (Figure 1A). We used hierarchical clustering of lead variant effect estimates to group loci and phenotypes. The variant effects largely correlated among VAR and CRS as one group, and CDTA and PA as another. This also distinguished broadly shared impact among VAR, CRS and NP in four loci (2q12.1, 5q22.1, 9p24.1, 10p14b). The CDTA-associated locus 2q33.3 showed highly concordant impact among VAR, CRS and NP as well. In total, 13 of 24 non-HLA loci had a co-directional association ($p < 0.00027$) with at least one other IURD phenotype.**”

Then we proceed with defining the cross-trait groups (lines 114–122, p. 6):

“**Genome-wide correlation analysis distinguished three IURD phenotype clusters from the GWAS results (Figure 1B).** We analyzed genetic correlation using LD Score regression⁵³. A high genetic correlation ($r_g > 75\%$) distinguished two clusters: I) VAR, CRS, NP, and NSD ($r_g \geq 78\%$); II) CDTA and PA ($r_g = 79\%$). Using a threshold of $r_g > 90\%$ further distinguished a genetically linked subgroup of known comorbid disorders²⁶: III) VAR, CRS, NP. We denoted these IURD groups as “sinonasal diseases” (I), “pharyngeal diseases” (II) and “chronic inflammatory sinonasal diseases” [CISDs, (III), Figure 2]. **CRNP had high genetic correlation with both pharyngeal diseases ($r_g \geq 67\%$) and VAR and CRS ($r_g \geq 87\%$), but not NP ($p = 0.051$), and was left outside these clusters.**”

Then we present results of combined trait analyses (lines 123–138, p. 6–7):

“**Cross-trait analysis using these IURD clusters identified six additional MTS loci. We performed cross-trait GWASs of sinonasal diseases (n = 25,235, Supplementary Table 2, Supplementary Figure 2A), pharyngeal diseases (n = 33,157, Supplementary Table 3, Supplementary Figure 2B), and CISD (n = 19,901, Supplementary Table 4, Supplementary Figure 2C) using the same GWAS pipeline as described above. In**

addition, we performed a GWAS of cases with any IURD ($n = 61,197$, Supplementary Table 5, Supplementary Figure 3). The genome-wide significant (GWS, $p < 5e-08$) VAR-associated locus 9q33.3 (near NEK6) was MTS associated with all IURDs [OR = 0.95 (0.93–0.97), $p = 1.75e-10$]. Similarly, the GWS CDTA-associated loci 1p36.23 and 16p11.2 were MTS associated with pharyngeal diseases, and the GWS novel NP-associated locus 2q22.3 near ZEB2 was MTS associated with sinonasal diseases. The GWAS of CISD identified the 15q22.33 locus near SMAD3 [OR = 1.08 (1.05–1.11), $p = 9.38e-10$], previously associated⁴⁵ with allergic disease, that was not observed in GWASs of VAR, CRS or NP. GWAS of sinonasal diseases additionally identified the locus 14q31.1 near NRXN3 [OR = 1.13 (1.08–1.18), $p = 3.47e-09$], not detected by CISD or NSD GWASs and not previously implicated. The six additional loci from cross-trait analyses brought our yield to 31 IURD-associated non-HLA loci.

An added analysis step using MTAG is presented to further the cross-trait analysis (lines 139–149, p. 7):

“To provide further robustness of our cross-trait analyses, we performed Multi-Trait Analysis of GWAS (MTAG⁵⁴, see Methods). Performing MTAG on all IURD traits supported three (near SMAD3, IL7R, and IKZF3) of the seven loci observed in the cross-trait analysis above (Supplementary Table 6). Among the IURD traits with no MTS loci (CRNP, NSD and CLT), MTAG supported CRNP association for four loci, of which 9q33.3 near NEK6 [OR = 0.99 (0.98–1.00)] replicated ($p = 0.0081$) in UKB (see below). NSD was associated with eight loci detected in other sinonasal GWASs. No new associations with CLT were observed. MTAG analysis additionally identified four novel GWS loci not seen in the original GWASs, of which one replicated ($p = 0.028$) in UKB: the 11q12.2 locus near FADS2 associated with NP [OR = 0.98 (0.97–0.99), $p = 2.7e-8$]. Together with the 31 independent MTS loci from IURD and cross-trait GWASs, the 11q12.2 locus brought our yield to 32 genomic loci.

We then present the Bayesian framework analysis, which is unchanged save for the previous Figures 3 and 4 now being presented as a single Figure 3. After our initial associations, we perform a meta-analysis with newly designed endpoints in UKB (lines 177–203, p. 9–10):

“Meta-analysis with UKB supported eight additional GWS loci, to a grand total of 40 non-HLA loci (Supplementary Table 7). For replication, we analyzed association of lead variants of all 58 GWS non-HLA loci in the UK Biobank (Supplementary Tables 1 and 8). We mapped the IURD and oral phenotypes to corresponding UKB read codes (Methods), and meta-analyzed variants with codirectional effects (38 loci of 58) between the cohorts. Meta-analysis resulted in GWS association for 29 of 38 codirectional loci (Table 2). There remained eight loci with MTS association in FinnGen, and three loci which replicated with a codirectional impact ($p < 0.05$) in UKB that were not GWS in meta-analysis (Table 3). All five VAR associations replicated in UKB including the novel NEK6 locus, albeit at a significantly ($p_z = 0.0020$) milder impact. Two CRS-associated variants at loci 5q22.1 and 9p24.1 showed significant codirectional impact in UKB at $p < 0.01$, with four other CRS-associated non-HLA loci not replicating despite 2q12.1 being previously³⁹ linked with NP. In addition to the eleven NP-associated loci overlapping ten previously reported loci in UKB, we replicated the two novel NP loci 1q21.3 and 2q22.3.

“Most loci linked to pharyngeal diseases showed high concordance in the UKB analysis despite significantly lower case counts. CDTA and PA had far lower effective sample sizes in UKB compared to FinnGen (4.6 % for CDTA, 13.3 % for PA), and a single-variant CDTA locus (rs774674736-D at 19q13.43) has not been genotyped in UKB. Lead variants of 13 of the 19 remaining CDTA-associated non-HLA loci showed codirectional and similar impact ($p_z > 0.05$) between UKB and FinnGen, and ten of the 13 loci were GWS in meta-analysis (Table 2, Supplementary Table 1). Five other MTS associated CDTA loci showed counter-directional effects in UKB despite adequate power ($> 70\%$), including the high-impact ($OR_{fg} = 1.43$) 17p11.2 locus near TNFRSF13B, previously associated with tonsillectomy in 23andMe⁴⁰ (Table 3). The PA-associated 3q21.2 locus also replicated [$OR_{ukb} = 0.81$, $p_{ukb} = 0.00054$], and along with loci 3q12.3 and 13q21.33 were GWS in meta-analysis with UKB. Statistical power for replication was $< 80\%$ for seven of the CDTA lead variants. However, as only one of these did not replicate, failure to replicate is more likely linked to the non-representative case count in UKB ($n = 1180$), differences in LD structure, and Finnish-specific low-frequency variants.

The 29 meta-analyzed loci are now presented in Table 2, and 11 loci with either MTS or nominal replication are presented in Table 3. Supplementary Table 7 has also been added to clarify and summarize the line of evidence considered behind each locus.

In the Discussion, the new results have led to the following changes:

Lines 332–336, p. 17:

“To understand the genetic predisposition landscape of infectious and inflammatory upper respiratory diseases, we genome-wide analyzed these diseases both individually and in groups and individually. In total, we identified **41** loci, of which **twelve** have not been previously reported to associate with any of the IURDs. Among the **41** loci, our fine-mapped credible sets identified **nine** coding variants, ~~including four that are enriched in the Finnish population.~~

Lines 347–354, p. 17–18:

Of the **40 reported non-HLA** loci, **17** were uniquely observed in a single IURD phenotype GWAS. The remaining **23** loci had highly similar odds ratios in two or more phenotypes, even if the signal did not reach the genome-wide significance threshold in all diseases. This is in line with previous epidemiological and histopathological evidence that highlights links between inflammatory sinonasal diseases^{5,6,26,42}. Similarly, pharyngeal diseases associate with **eleven** loci previously linked to tonsillectomy^{40,41} as well as **one locus** associated with self-reported strep throat and childhood ear infections⁴⁰. While there is a shared genetic contribution for **three** loci to all IURDs[...]

The following changes have also been made to the Methods section:

Lines 465–467 (p. 23):

“We analyzed genome-wide association of cases of **eight** IURDs (Table 1; total $n = 61,197$) against 199,208 controls with no IURDs in the FinnGen dataset.”

Lines 470–474 (p. 23):

“**Two oral phenotypes, DPPT and CP, were separately analyzed due to epidemiological**

overlap with pharyngeal and sinonasal IURDs. The study-wide level of significance (multiple testing significant, MTS) was set at $5e-9$ to correct for the simultaneous analysis of ten different diseases. Non-MTS GWS loci ($5e-9 < p < 5e-8$) were considered significant only if they replicated in UKB or meta-analysis.

Lines 509–528 (p. 25–26):

“For replication, we analyzed association of lead variants of all 59 GWS non-HLA loci in the UK Biobank (Supplementary Table 1). We mapped the IURD and oral endpoints in FinnGen to corresponding UKB read codes (Supplementary Table 15) using ICD-to-Phecode mapping, in addition to manual curation of these codes based on description, using hospital data, cause of death registry, and for a subset ($n = 230,000$) also GP data. UKB variants were aligned to variants in FinnGen. In case of no exact match between SNPs (ref and alt differ between studies), matching was tried by flipping strand and/or switching ref->alt and alt->ref for the UKB variant. Variants were tested against the constructed endpoints in the UKB European population using logistic regression. Covariates were reported gender, baseline age, and PCs 1–10. We used as controls all UKB participants with no identified IURD or oral endpoint ($n = 298,846$). To estimate heterogeneity of effect between the cohorts, a test statistic was calculated with the formula

$$z = (\beta_1 - \beta_2)^2 / (SE_1^2 + SE_2^2)$$

where β_i is the effect size of study i , and SE_i is the standard error of the effect estimate in study i . The test statistic z was assumed to follow a χ^2 distribution with one degree of freedom. Variants with a $p_z < 0.05$ were considered heterogeneous. Only co-directional variants were meta-analyzed. We used inverse-variance weighted meta-analysis under a fixed-effect assumption. Variants were considered significant if they had a GWS impact after meta-analysis with UKB, had a GWS association in FinnGen and replicated a codirectional association ($p < 0.05$) in UKB, or MTS impact in FinnGen alone.”

- *I understand the work of the authors which is somewhat methodological when grouping all the IURDs phenotypes, but I feel that this demonstration somewhat limits the conclusions and the robustness of the results. If the authors succeed to have a clearer presentation including replication and combined analysis with abroad. A large number of the results presented here might be false positive, since 1) the cutoff used $5E-8$ had been set under different circumstances, population and number of markers 2) Many of the claimed association are less than an order of magnitude from the lenient threshold 3) most of them 24 out of 41 do not replicate nominally in their foreign group.*
 - o As stated above, we have now reordered the manuscript, which hopefully presents the results more clearly. As also presented above, in order to make our results more robust, we have now applied a threshold of multiple testing significance (MTS) based on the ten independent phenotypes. Additionally, the UKB meta-analysis has been redone using redesigned endpoints with combined GP data (instead of PanUK).
- *Additionally, one would expect that the authors attempt replication of all the reported variants in the literature for the phenotypes they study, in order to assess how comparable their groups*

are with the one used for previously reported associations. For example, a variant in ALOX15 has been reported to confer a strong and significant protection against NP and CRS. It is somewhat unexpected not to see it mentioned in this manuscript.

- We agree, and have now gathered a separate table (Supplementary Table 9) reporting previous associations. We have additionally added the following summarizing paragraph to the Results section (lines 204–213, p. 10):

“We also investigated the association of previously reported GWAS of similar traits (Supplementary Table 9). We identified as ‘replicated’ any previously reported locus with a similar directional OR and $p < 0.01$ in our analyses. In this way, our VAR GWAS replicated 18 of the previously reported^{41,44-48} 34 allergic rhinitis loci with lead variants genotyped in FinnGen. Similarly, all 10 loci previously associated³⁹ with CRS and NP were replicated in our respective GWASs. This included the protective missense variant rs34210653-A in ALOX15, associated with NP [OR = 0.52 (0.38–0.71), $p = 1.62E-05$] and CRS [OR = 0.81 (0.67–0.97), $p = 0.016$]. For the 26 of 35 tonsillectomy-associated loci genotyped in FinnGen, our CDTA and PA analyses replicated 22. Finally, 2 of 2 loci associated with strep throat⁴⁰ also replicated in our CDTA and PA analyses.

A GWAS meta-analysis on allergic rhinitis (by Waage et al 2018) was also added to the Introduction, with the following edits (lines 68–75, p. 3):

“Previous genetic studies of **non-allergic** IURDs and related immune responses have largely focused on rare variants³⁶ and the HLA region^{37,38}. IURD-related GWAS have been reported of CRS and NP³⁹, tonsillectomy and childhood ear infections^{40,41}, cold sores, mononucleosis, strep throat, pneumonia and myringotomy⁴⁰, **hay fever**^{39,42} and of infective diseases caused by specific airway-related microbes such as pneumococcus⁴² and staphylococcus aureus⁴³. **The common variant burden of allergic diseases such as AR have been more extensively studied**^{41,44-48}. However, no prior research has analyzed shared genetic contributions of IURDs.

- *One of the main issue with this manuscript is the phenotype. I understand the angle the authors have taken: starting with a combination of phenotypes in Finland. But I truly am of the opinion that it makes it very unclear what is really claimed*

I would strongly suggest to start with specific phenotypes, performing meta-analysis with other population than Finland and with more stringent significance cutoffs to then arrive to a set of robust associations that they can then assess in the other subsets. In the UK biobank some of these diagnosis have to be derived from the GP sets in addition to hospital diagnosis

- We thank for this comment, and feel this is resolved by the edits described above.
- *Also the authors might realize that a lot of these diagnosis are differential diagnosis for each other’s, and that a large number of the patients probably overlap between subsets.*
 - This is good point. In our initial submission, we sought to highlight this potential bias with a broad note in the limitations section (p. 20–21): “Second, the IURD phenotypes

are diagnosed by specialists, often in hospital setting, and thus likely quite accurate but are therefore subject to ascertainment bias (collider bias) with other disorders – a feature of study design that can inflate correlation estimates with other diseases.”

However, with the point raised by Reviewer #1 we concede that this treatment might not quite address what the Reviewer is asking. In general, the overlap of even known similar diagnoses is not here complete. To highlight the correlation, we have now prepared Supplementary Table 13 to show the co-occurrence of IURD endpoints. We have also added the following sentence to Methods (lines 442–443, p. 22):

“The co-occurrence of the IURD diagnoses is summarized in Supplementary Table 13.”

- *Personally, I recommend to focus on the strong signal, choose more significant cutoffs. And present the results that are specific on the subsets, once combined with public datasets.*

As mentioned in different places above, the cutoff for statistical significance of $5E-8$ has been proposed many years back at the time where sample size and number of markers tested were different. From the attempt of replication currently displayed it is possible that a large fraction of the association are not true positive. If they were, the ones very close to the cutoff are likely suffering from winner’s curse, i.e. an inflation of effect

- We believe these concerns are met by the multiple testing cutoff, the restructuring that focuses on phenotype-specific analyses, and the new UKB replication.
- *Using meta-analysis with foreign data should be performed when available.*
 - This a good point and has now been attempted to be met with reporting inverse variance-weighted meta-analyzed effects of FinnGen and UKB results in the supplement. An inherent feature of genetic association analysis in population isolates is that loci identified with population-specific enriched variants are hard or impossible to be analyzed adequately in more mixed populations. In these cases a non-replication does not necessarily mean false positive.
- *Again, I would rather have claims specific to the subsets that are robust presented first, with combined significance with the foreign groups.*
- *I would present replications of all previous reported markers for these phenotypes.*
 - We again agree, and thank Reviewer #1 for these valuable comments that we feel have allowed us to improve our manuscript substantially.

Reviewer #2 comments:

The manuscript by Saarentaus and colleagues describes a GWAS of inflammatory and infectious upper respiratory disease (IURD) in ~260k FinnGen study participants. They identify 59 genome-wide significant loci associated with this heterogeneous disease definition and/or component diseases of the umbrella term. I have a few comments for consideration by the authors.

- Identifying loci by lumping together different diseases into a single ‘mega case’ group is a reasonable approach but I wonder if there are better alternatives. Have the authors explored using Genomic SEM (or similar approaches) to identify genetic factors that are shared across the individual disease GWAS?
 - We recognize that traits with partially shared genetic etiologies can be analyzed in several ways. The paper presentation has now been reordered, such that subphenotypes results are presented first (as detailed below). In order to make our cross-trait analyses more robust, and provide further support for our approach, we have now employed Multi-Trait Analysis of GWAS (MTAG), which replicates three of 17 loci detected in cross-trait analyses. MTAG also highlighted four new loci, one of which also replicated in our new UKB analysis and is consequently separately reported. The limitation of MTAG is a reliance on AF > 1 % variants, so not even all loci from the original GWAS are seen in this approach. Therefore, not seeing support for some new loci from the cross-trait analysis is not considered as evidence against them. The following additions have been made to the manuscript and hope that it addresses this reviewer’s concern:

Results (lines 139–149, p. 7):

“To provide further robustness of our cross-trait analyses, we performed Multi-Trait Analysis of GWAS (MTAG⁵⁴, see Methods). Performing MTAG on all IURD traits supported three (near SMAD3, IL7R, and IKZF3) of the seven loci observed in the cross-trait analysis above (Supplementary Table 6). Among the IURD traits with no MTS loci (CRNP, NSD and CLT), MTAG supported CRNP association for four loci, of which 9q33.3 near NEK6 [OR = 0.99 (0.98–1.00)] replicated ($p = 0.0081$) in UKB (see below). NSD was associated with eight loci detected in other sinonasal GWASs. No new associations with CLT were observed. MTAG analysis additionally identified four novel GWS loci not seen in the original GWASs, of which one replicated ($p = 0.028$) in UKB: the 11q12.2 locus near FADS2 associated with NP [OR = 0.98 (0.97–0.99), $p = 2.7e-8$]. Together with the 31 independent MTS loci from IURD and cross-trait GWASs, the 11q12.2 locus brought our yield to 32 genomic loci.

Methods (lines 482–485, p. 24):

“We performed Multi-Trait Analysis of GWAS (MTAG)⁵⁴ using GWAS summary statistics from all IURD phenotype GWASs jointly. Only variants with MAF > 1 % were considered ($n = 6,868,381$). As the approach has an elevated type II error rate, novel loci identified by MTAG were only considered meaningful if replicated in the UKB analysis.

- The authors have effectively run 12 GWASs but corrected for only one ($P < 5 \times 10^{-8}$) from what I can see. They may want to justify this approach.

- As this issue was raised also by Reviewer #1 and highlighted by the editor, we have now applied a threshold of multiple testing significance (MTS) based on the ten independent phenotypes, in addition to improving our UKB meta-analysis with new endpoints using GP data (instead of PanUK).

In order to present our results more appropriately, we have added major revisions to restructure the Results section of the manuscript. The new version first presents the specific phenotype analyses (nasal polyps, peritonsillar abscess), then investigates the shared heritability through genetic correlation and cross-trait analyses. We then present the Bayesian analysis pipeline to disentangle shared and distinct loci. We have also designed and performed our endpoints in the UKBB from read codes (incl GP data). In this way, we now report only loci that are either genome-wide significant in meta-analysis, multiple testing significant ($5e-9$ for ten independent phenotypes), or genome-wide significant with nominal replication in UKB.

The replication chapter (lines 177–203, p. 9–10) now reads:

“Meta-analysis with UKB supported eight additional GWS loci, to a grand total of 40 non-HLA loci (Supplementary Table 7). For replication, we analyzed association of lead variants of all 58 GWS non-HLA loci in the UK Biobank (Supplementary Tables 1 and 8). We mapped the IURD and oral phenotypes to corresponding UKB read codes (Methods), and meta-analyzed variants with codirectional effects (38 loci of 58) between the cohorts. Meta-analysis resulted in GWS association for 29 of 38 codirectional loci (Table 2). There remained eight loci with MTS association in FinnGen, and three loci which replicated with a codirectional impact ($p < 0.05$) in UKB that were not GWS in meta-analysis (Table 3). All five VAR associations replicated in UKB including the novel NEK6 locus, albeit at a significantly ($p_z = 0.0020$) milder impact. Two CRS-associated variants at loci 5q22.1 and 9p24.1 showed significant codirectional impact in UKB at $p < 0.01$, with four other CRS-associated non-HLA loci not replicating despite 2q12.1 being previously³⁹ linked with NP. In addition to the eleven NP-associated loci overlapping ten previously reported loci in UKB, we replicated the two novel NP loci 1q21.3 and 2q22.3.

“Most loci linked to pharyngeal diseases showed high concordance in the UKB analysis despite significantly lower case counts. CDTA and PA had far lower effective sample sizes in UKB compared to FinnGen (4.6 % for CDTA, 13.3 % for PA), and a single-variant CDTA locus (rs774674736-D at 19q13.43) has not been genotyped in UKB. Lead variants of 13 of the 19 remaining CDTA-associated non-HLA loci showed codirectional and similar impact ($p_z > 0.05$) between UKB and FinnGen, and ten of the 13 loci were GWS in meta-analysis (Table 2, Supplementary Table 1). Five other MTS associated CDTA loci showed counter-directional effects in UKB despite adequate power ($> 70\%$), including the high-impact ($OR_{fg} = 1.43$) 17p11.2 locus near TNFRSF13B, previously associated with tonsillectomy in 23andMe⁴⁰ (Table 3). The PA-associated 3q21.2 locus also replicated [$OR_{ukb} = 0.81$, $p_{ukb} = 0.00054$], and along with loci 3q12.3 and 13q21.33 were GWS in meta-analysis with UKB. Statistical power for replication was $< 80\%$ for seven of the CDTA lead variants. However, as only one of these did not replicate, failure to replicate is more likely linked to the non-representative case count in UKB ($n = 1180$), differences in LD structure, and Finnish-specific low-frequency variants.

Methods:

Lines 509–528 (p. 25–26):

“For replication, we analyzed association of lead variants of all 59 GWS non-HLA loci in the UK Biobank (Supplementary Table 1). We mapped the IURD and oral endpoints in FinnGen to corresponding UKB read codes (Supplementary Table 15) using ICD-to-Phecode mapping, in addition to manual curation of these codes based on description, using hospital data, cause of death registry, and for a subset (n = 230,000) also GP data. UKB variants were aligned to variants in FinnGen. In case of no exact match between SNPs (ref and alt differ between studies), matching was tried by flipping strand and/or switching ref->alt and alt->ref for the UKB variant. Variants were tested against the constructed endpoints in the UKB European population using logistic regression. Covariates were reported gender, baseline age, and PCs 1–10. We used as controls all UKB participants with no identified IURD or oral endpoint (n = 298,846). To estimate heterogeneity of effect between the cohorts, a test statistic was calculated with the formula

$$z=(\beta_1-\beta_2)^2/(SE_1^2+SE_2^2)$$

where β_i is the effect size of study i , and SE_i is the standard error of the effect estimate in study i . The test statistic z was assumed to follow a χ^2 distribution with one degree of freedom. Variants with a $p_z < 0.05$ were considered heterogeneous. Only co-directional variants were meta-analyzed. We used inverse-variance weighted meta-analysis under a fixed-effect assumption. Variants were considered significant if they had a GWS impact after meta-analysis with UKB, had a GWS association in FinnGen and replicated a codirectional association ($p < 0.05$) in UKB, or MTS impact in FinnGen alone.”

- “Lead variants of the 58 non-HLA GWS loci were associated with 2,861 endpoints in FinnGen” – I suspect this number would be substantially lower if co-localization analyses were run. Presumably it’s possible that many of these associations are in “LD shadows” of other signals?
 - o This is a good point, and part of the reason these results were only considered in the advent of recurring PheWAS hits. We have now incorporated a co-localization analysis similar to eCAVIAR to provide more robust evidence for shared causal variants. These results (Supplementary Table 12) have replaced the supplementary PheWAS table (previously Table S7) and we have replaced sections of the PheWAS results paragraph as follows (lines 293–308, p. 14–15):

“To evaluate shared impact to non-IURD phenotypes, we also here used an in-house pipeline based on eCAVIAR⁷² to evaluate colocalization of IURD loci and 2,861 endpoints in the FinnGen PheWeb. We linked 21 non-HLA IURD loci to 380 credible sets from 108 FinnGen endpoints (Supplementary Table 12). Causal posterior probability was > 80 % for colocalization between the NP locus 5q22.1 and asthma endpoints, and between the CDTA locus 12p13.31a and acute appendicitis. Beyond these, causal posterior agreement was > 60 % for nine IURD loci and 54 non-IURD phenotypes. These 54 phenotypes included infectious and inflammatory disorders of the upper respiratory tract that were not included in our definition of IURD: acute

sinusitis (9p24.1 near IL33) and non-suppurative otitis media (17p11.2 near TNFRSF13B). Asthma endpoints colocalized with six loci associated with NP, CRS, and VAR – similar to previously reported loci linked with CRS with NP (CRSwNP)³⁹. Phenome-wide association to autoimmune diseases was observed near RAB5B, HORMAD2 and NEK6, and to inflammatory bowel disease near ARNT, CDC42SE2, and GATA3. The CDTA-associated locus near ABO was associated with pulmonary heart disease, duodenal ulcer, and deep vein thrombosis. Broad colocalization was also observed for the VAR locus 11q13.5 (near EMSY), which colocalized with atopic dermatitis, conjunctivitis, and inflammatory bowel diseases in addition to asthma.

The pipeline is described in Methods (p. 27, lines 537–559), replacing the previous PheWAS chapter:

“We used an in-house pipeline based on eCAVIAR⁷² to evaluate colocalization with GWAS summary statistics. The pipeline uses SuSiE-finemapped CSs as inputs, and calculates a causal posterior probability (CLPP) that the same locus is causal in both studies. CLPP is defined as the sum of the products of SuSiE-finemapped posterior probabilities (PIP; x for phenotype 1, y for phenotype 2) for each variant i shared in credible sets of both phenotypes, such that for credible set k:

$$[CLPP]_k = \sum_{i \in CS} [x_i * y_i]$$

CLPP was considered significant if it was higher than 20 %. Another colocalization metric, causal posterior agreement (CLPA), was devised as a metric independent of CS size. CLPA represents the agreement between the fine-mapping results in both studies, and is defined as the sum of minimum PIP of shared variants between CSs from phenotype 1 and 2. CLPA was considered significant if higher than 50 %. For gene expression, the GWAS summary statistics were derived from GTEx v8⁷³ and EBI⁷⁴. Phenotype summary statistics were derived from the FinnGen PheWeb.

- *“We then compared SNPs with expression data from GTEx v8 and DICE, which showed change in the expression of a total of 264 genes in all 58 non-HLA loci in total” – this statement really highlights the limitations of using tools like FUMA to assign causal genes to GWAS hits. The mapping of non-synonymous variants appeared methodologically robust, and I’d suggest the authors apply a similarly conservative approach to eQTL mapping. At a minimum the authors need to ensure they use eQTL integration approaches that reduce spurious associations due to more coincidental LD overlaps.*

- o We agree, and have now excluded FUMA from our analysis pipeline. We have instead incorporated the same co-localization pipeline as detailed above. Supplementary Table 10 (Coloc eQTL results) now presents these results, which are also described in the Results (p. 13, lines 265–276):

“Next, we used an in-house pipeline based on eCAVIAR⁷² to evaluate the impact on gene expression (Methods). In brief, we colocalized fine-mapped credible sets of IURD GWAS summary statistics with similarly fine-mapped credible sets of eQTLs in GTEx v8⁷³ and EBI⁷⁴ eQTL databases. We linked 2,129 eQTL credible sets overlapping IURD credible sets from 27 loci (Supplementary Table 10). 8 IURD loci paired with 427 eQTLs in tissues outside the CNS and gonads with a minimum 60 % posterior agreement. Three loci had at least 80 % causal posterior agreement with credible sets of eQTLs in

immunological cell types. The CDTA-associated ($OR_{fg} = 1.10$) 2q13 locus decreased expression of *MIR4435-2HG* in lymphoblastoid cells and CD14+ CD16- classical monocytes. The CDTA-associated ($OR_{fg} = 1.16$) 12p13.31 increased *LTBR* expression in macrophages, CD14+ CD16- classical monocytes, lymphoblastoid cells and T cells. The NP-associated ($OR_{meta} = 1.15$) locus 12q13.2 associated with increased *RAB5B* expression in CD4+ $\alpha\beta$ -T-cells and neutrophils.

The Discussion has also seen the following edits:

Lines 373–377 , p. 19:

“Pharyngeal diseases implicate ~~many~~ genes linked with immune deficiency. Non-synonymous variants were implicated in **eight** loci, highlighting **three** genes linked with immune deficiency and immune-mediated disorders. Interestingly, in **two of these** genes (NFKB1, ~~CARMIL2~~, and IL7R) with previously established risk variants for immune deficiency¹⁵, we identify non-synonymous variants with decreased risk for IURDs. ~~Differing expression linked two additional immune deficiency-linked genes (IKBKB with CDTA and STAT2 with NP), thus linking eleven immunomodulatory genes to IURDs in total.~~”

Lines 381–382 , p. 19:

“**Links to** immune-mediated disorders, such as asthma and inflammatory bowel diseases, **were also observed in colocalization analysis.**”

Lines 395–396 , p. 20:

“**Phenome-wide colocalization analysis** of the *ABO* gene cluster [...]”

The characterization of loci Methods section now reads (lines 530–535, p.26):

“**After the initial detection of genome-wide significant (GWS; $p < 5e-8$) associated SNPs, we chose lead SNPs based on lowest p-value, and GWS SNPs in the same locus were grouped based on genomic distance $< 2Mb$, $r^2 > 0.1$ with lead SNP. We used SuSiE⁶⁰ for detection of credible sets of causal variants, with a Finnish-based reference panel [Sequencing Initiative Suomi⁹¹] for LD structure and imputation. Only credible sets with at least one genome-wide significant (GWS; $p < 5e-8$) SNP were considered.**”

- **Impact**

Personally I feel this manuscript lacks the novel biological or epidemiological insights that warrant publication in journal such as Nature Genetics. A more incremental but rigorous study of this type may be better suited to Communications Biology or Nature Communications

- **Reproducibility**

From what I can see it is not possible for anyone outside of FinnGen to access individual level data and reproduce these results. This is a shame given the precedent set by studies such as UK Biobank.

We couldn't agree more with the frustration of this reviewer. The current European regulatory environment and additional national regulations, some of which are just becoming stricter, strongly opposes the open science goals, by preventing wide sharing of individual level health and genome data.

UK Biobank was initiated with a consent (and clearly very smart lawyers) that enables individual level sharing. Now as UK is not in EU anymore, they do not need to be restricted by GDPR or other EU restrictions. That said, we are doing our best to be as open as we can within this regulatory environment. The next best alternative that we can apply is (1) that we make summary statistics, PheWAS and colocalization results available for all qualified researchers and research institutions (https://www.finnngen.fi/en/access_results). (2) The FinnGen team is very open and excited to collaborate with non-FinnGen partners if individual level data is necessary for the analysis. The number of partner institutions in FinnGen is so large that it should not be very difficult to find a collaborator that is interested in a given phenotype.

Reviewer #3 comments:

The authors performed genome-wide association studies of a series of inflammatory and infectious upper respiratory diseases and identified 59 loci associated with one or more of these diseases, of which 23 are novel. They identified a high degree of sharing between many IURD subtypes, as well as with other related phenotypes such as asthma and allergies. The analyses are appropriate and well-performed and the paper is well-written. I have some questions and concerns that I hope the authors will be able to address:

- On page 3, several previous genetic studies of IURDs are mentioned, but the authors do not seem to place their new results in the context of these studies. For instance, when compared to Kristjansson et al 2019, the current manuscript has twice the number of CRS cases and a similar number of NP cases, yet did not report associations at the loss-of-function variant in ALOX15. I would be curious to know whether these and other variants reported in the literature also showed associations in FinnGen, and if not, whether aspects of study design (e.g. case definition) or differences in LD patterns can lead to certain loci not being detected
 - o We agree, this was clearly a mistake in the previous version of the manuscript. We have now gathered a separate table (Supplementary Table 9) reporting previous associations. We have additionally added the following summarizing paragraph to the Results section (lines 204–213, p. 10):

“We also investigated the association of previously reported GWAS of similar traits (Supplementary Table 9). We identified as ‘replicated’ any previously reported locus with a similar directional OR and $p < 0.01$ in our analyses. In this way, our VAR GWAS replicated 18 of the previously reported^{41,44-48} 34 allergic rhinitis loci with lead variants genotyped in FinnGen. Similarly, all 10 loci previously associated³⁹ with CRS and NP were replicated in our respective GWASs. This included the protective missense variant rs34210653-A in ALOX15, associated with NP [OR = 0.52 (0.38–0.71), $p = 1.62E-05$] and CRS [OR = 0.81 (0.67–0.97), $p = 0.016$]. For the 26 of 35 tonsillectomy-associated loci genotyped in FinnGen, our CDTA and PA analyses replicated 22. Finally, 2 of 2 loci associated with strep throat⁴⁰ also replicated in our CDTA and PA analyses.

A GWAS meta-analysis on allergic rhinitis (by Waage et al 2018) was also added to the Introduction, with the following edits (lines 68–75, p. 3):

“Previous genetic studies of **non-allergic** IURDs and related immune responses have largely focused on rare variants³⁶ and the HLA region^{37,38}. IURD-related GWAS have been reported of CRS and NP³⁹, tonsillectomy and childhood ear infections^{40,41}, cold sores, mononucleosis, strep throat, pneumonia and myringotomy⁴⁰, **hay fever**^{39,42} and of infective diseases caused by specific airway-related microbes such as pneumococcus⁴² and staphylococcus aureus⁴³. **The common variant burden of allergic diseases such as AR have been more extensively studied**^{41,44-48}. However, no prior research has analyzed shared genetic contributions of IURDs.

- With regards to the replication work in UKB:
 - o Page 7: “We found that the effect size differed ($p < 0.05$) only for the CRS association to the locus 5q22.1 (near WDR36) and the NP association to the locus 1q211.3 (near ARNT).”

In Table S6, it looks like the GAS2L2/TAF15 association with CRS was also not replicated and has a significant effect size difference in UKB

- We apologize for missing this, and have corrected this in our new version of the replication and meta-analysis that now reads (lines 177–203, p. 9–10):

“Meta-analysis with UKB supported eight additional GWS loci, to a grand total of 40 non-HLA loci (Supplementary Table 7). For replication, we analyzed association of lead variants of all 58 GWS non-HLA loci in the UK Biobank (Supplementary Tables 1 and 8). We mapped the IURD and oral phenotypes to corresponding UKB read codes (Methods), and meta-analyzed variants with codirectional effects (38 loci of 58) between the cohorts. Meta-analysis resulted in GWS association for 29 of 38 co-directional loci (Table 2). There remained eight loci with MTS association in FinnGen, and three loci which replicated with a codirectional impact ($p < 0.05$) in UKB that were not GWS in meta-analysis (Table 3). All five VAR associations replicated in UKB including the novel NEK6 locus, albeit at a significantly ($p_z = 0.0020$) milder impact. Two CRS-associated variants at loci 5q22.1 and 9p24.1 showed significant codirectional impact in UKB at $p < 0.01$, with four other CRS-associated non-HLA loci not replicating despite 2q12.1 being previously³⁹ linked with NP. In addition to the eleven NP-associated loci overlapping ten previously reported loci in UKB, we replicated the two novel NP loci 1q21.3 and 2q22.3.

“Most loci linked to pharyngeal diseases showed high concordance in the UKB analysis despite significantly lower case counts. CDTA and PA had far lower effective sample sizes in UKB compared to FinnGen (4.6 % for CDTA, 13.3 % for PA), and a single-variant CDTA locus (rs774674736-D at 19q13.43) has not been genotyped in UKB. Lead variants of 13 of the 19 remaining CDTA-associated non-HLA loci showed codirectional and similar impact ($p_z > 0.05$) between UKB and FinnGen, and ten of the 13 loci were GWS in meta-analysis (Table 2, Supplementary Table 1). Five other MTS associated CDTA loci showed counter-directional effects in UKB despite adequate power ($> 70\%$), including the high-impact ($OR_{fg} = 1.43$) 17p11.2 locus near TNFRSF13B, previously associated with tonsillectomy in 23andMe⁴⁰ (Table 3). The PA-associated 3q21.2 locus also replicated [$OR_{ukb} = 0.81$, $p_{ukb} = 0.00054$], and along with loci 3q12.3 and 13q21.33 were GWS in meta-analysis with UKB. Statistical power for replication was $< 80\%$ for seven of the CDTA lead variants. However, as only one of these did not replicate, failure to replicate is more likely linked to the non-representative case count in UKB ($n = 1180$), differences in LD structure, and Finnish-specific low-frequency variants.

Methods:

Lines 509–528 (p. 25–26):

“For replication, we analyzed association of lead variants of all 59 GWS non-HLA loci in the UK Biobank (Supplementary Table 1). We mapped the IURD and oral endpoints in FinnGen to corresponding UKB read codes (Supplementary Table 15) using ICD-to-Phecode mapping, in addition to manual curation of these codes based on description, using hospital data,

cause of death registry, and for a subset (n = 230,000) also GP data. UKB variants were aligned to variants in FinnGen. In case of no exact match between SNPs (ref and alt differ between studies), matching was tried by flipping strand and/or switching ref->alt and alt->ref for the UKB variant. Variants were tested against the constructed endpoints in the UKB European population using logistic regression. Covariates were reported gender, baseline age, and PCs 1–10. We used as controls all UKB participants with no identified IURD or oral endpoint (n = 298,846). To estimate heterogeneity of effect between the cohorts, a test statistic was calculated with the formula

$$z = (\beta_1 - \beta_2) / \sqrt{SE_1^2 + SE_2^2}$$

where β_i is the effect size of study i, and SE_i is the standard error of the effect estimate in study i. The test statistic z was assumed to follow a χ^2 distribution with one degree of freedom. Variants with a $p_z < 0.05$ were considered heterogeneous. Only co-directional variants were meta-analyzed. We used inverse-variance weighted meta-analysis under a fixed-effect assumption. Variants were considered significant if they had a GWS impact after meta-analysis with UKB, had a GWS association in FinnGen and replicated a codirectional association ($p < 0.05$) in UKB, or MTS impact in FinnGen alone.”

- *In general, how should we interpret loci that failed replication, despite having over 95% power to do so at $\alpha=0.05$? Could it be that some of these diseases that occurred earlier in life are not well-captured in the UKB’s cohort of middle-aged participants? Is there any evidence of different LD patterns between Finnish and UK population at some of the non-replicating loci?*

Similarly, in UKB, the CDTA phenotype had the smallest effective sample size, while in FinnGen, this phenotype had the largest. Is this a consequence of diseases that occur earlier in life not being as well captured in UKB? If that is the case, how should we interpret the replication results in UKB more generally?

- o We thank for these important insights. In discussion with Reviewer #1, we have now sought formal replication by enriching UK Biobank hospital-based endpoints together with GP data. In this context, the concerns raised by Reviewer #3 here also contributed to the following paragraph regarding UK Biobank replication in the results (lines 177–203, p. 9–10):
“Meta-analysis with UKB supported eight additional GWS loci, to a grand total of 40 non-HLA loci (Supplementary Table 7). For replication, we analyzed association of lead variants of all 58 GWS non-HLA loci in the UK Biobank (Supplementary Tables 1 and 8). We mapped the IURD and oral phenotypes to corresponding UKB read codes (Methods), and meta-analyzed variants with codirectional effects (38 loci of 58) between the cohorts. Meta-analysis resulted in GWS association for 29 of 38 co-directional loci (Table 2). There remained eight loci with MTS association in FinnGen, and three loci which replicated with a codirectional impact ($p < 0.05$) in UKB that were not GWS in meta-analysis (Table 3). All five VAR associations replicated in UKB including the novel NEK6 locus, albeit at a significantly ($p_z = 0.0020$) milder impact. Two CRS-associated variants at loci 5q22.1 and 9p24.1 showed significant

codirectional impact in UKB at $p < 0.01$, with four other CRS-associated non-HLA loci not replicating despite 2q12.1 being previously³⁹ linked with NP. In addition to the eleven NP-associated loci overlapping ten previously reported loci in UKB, we replicated the two novel NP loci 1q21.3 and 2q22.3.

“Most loci linked to pharyngeal diseases showed high concordance in the UKB analysis despite significantly lower case counts. CDTA and PA had far lower effective sample sizes in UKB compared to FinnGen (4.6 % for CDTA, 13.3 % for PA), and a single-variant CDTA locus (rs774674736-D at 19q13.43) has not been genotyped in UKB. Lead variants of 13 of the 19 remaining CDTA-associated non-HLA loci showed codirectional and similar impact ($p_z > 0.05$) between UKB and FinnGen, and ten of the 13 loci were GWS in meta-analysis (Table 2, Supplementary Table 1). Five other MTS associated CDTA loci showed counter-directional effects in UKB despite adequate power ($> 70\%$), including the high-impact ($OR_{ig} = 1.43$) 17p11.2 locus near TNFRSF13B, previously associated with tonsillectomy in 23andMe⁴⁰ (Table 3). The PA-associated 3q21.2 locus also replicated [$OR_{ukb} = 0.81$, $p_{ukb} = 0.00054$], and along with loci 3q12.3 and 13q21.33 were GWS in meta-analysis with UKB. Statistical power for replication was $< 80\%$ for seven of the CDTA lead variants. However, as only one of these did not replicate, failure to replicate is more likely linked to the non-representative case count in UKB ($n = 1180$), differences in LD structure, and Finnish-specific low-frequency variants.

In addition to the obvious possibility of the false positive signal, at least two possibilities exist. One is a difference in the signal recorded in the two study samples; the other concerns low-frequency/rare variants enriched in Finland, in a population isolate, that are extremely infrequent or absent in mixed populations and thus not reliably measurable outside of Finland.

These points have been added to the Discussion limitations paragraph (lines 419–425, p. 21):

“Finally, while the **VAR and NP analyses** in the **UKB** were well-powered for replication, the effective sample sizes for other **IURDs** were not sufficient for reliable replication analysis of many of the lead variants. **An inherent feature of genetic association analysis in population isolates is that loci identified with population-specific enriched variants are hard or impossible to be analyzed adequately in more mixed populations. A non-replication does not necessarily mean false positive.**

- *In the PheWAS analysis of lead SNPs, Sup table 7 shows 462 significant variant-phenotype pairs. How many of these pairs are actually being most likely driven by the same causal variant versus being driven by LD with another causal variant?*
 - This is a valid challenge. We have now incorporated a co-localization analysis similar to eCAVIAR to provide more robust evidence for shared causal variants. These results (Supplementary Table 12) have replaced the supplementary PheWAS table (previously Table S7) and we have replaced sections of the PheWAS results paragraph as follows (lines 293–308, p. 14–15):

“To evaluate shared impact to non-IURD phenotypes, we also here used an in-house pipeline based on eCAVIAR⁷² to evaluate colocalization of IURD loci and 2,861

endpoints in the FinnGen PheWeb. We linked 21 non-HLA IURD loci to 380 credible sets from 108 FinnGen endpoints (Supplementary Table 12). Causal posterior probability was > 80 % for colocalization between the NP locus 5q22.1 and asthma endpoints, and between the CDTA locus 12p13.31a and acute appendicitis. Beyond these, causal posterior agreement was > 60 % for nine IURD loci and 54 non-IURD phenotypes. These 54 phenotypes included infectious and inflammatory disorders of the upper respiratory tract that were not included in our definition of IURD: acute sinusitis (9p24.1 near IL33) and non-suppurative otitis media (17p11.2 near TNFRSF13B). Asthma endpoints colocalized with six loci associated with NP, CRS, and VAR – similar to previously reported loci linked with CRS with NP (CRSwNP)³⁹. Phenome-wide association to autoimmune diseases was observed near RAB5B, HORMAD2 and NEK6, and to inflammatory bowel disease near ARNT, CDC42SE2, and GATA3. The CDTA-associated locus near ABO was associated with pulmonary heart disease, duodenal ulcer, and deep vein thrombosis. Broad colocalization was also observed for the VAR locus 11q13.5 (near EMSY), which colocalized with atopic dermatitis, conjunctivitis, and inflammatory bowel diseases in addition to asthma.

The Discussion has also seen the following edits:

Lines 373–377 , p. 19:

“Pharyngeal diseases implicate ~~many~~ genes linked with immune deficiency. Non-synonymous variants were implicated in **eight** loci, highlighting **three** genes linked with immune deficiency and immune-mediated disorders. Interestingly, in **two of these** genes (NFKB1, ~~CARMIL2~~, and IL7R) with previously established risk variants for immune deficiency¹⁵, we identify non-synonymous variants with decreased risk for IURDs. ~~Differing expression linked two additional immune deficiency-linked genes (IKBKB with CDTA and STAT2 with NP), thus linking eleven immunomodulatory genes to IURDs in total.~~”

Lines 381–382 , p. 19:

“**Links to** immune-mediated disorders, such as asthma and inflammatory bowel diseases, **were also observed in colocalization analysis.**”

Lines 395–396 , p. 20:

“**Phenome-wide colocalization analysis** of the *ABO* gene cluster [...]”

The pipeline is described in Methods (p. 27, lines 537–559), replacing the previous PheWAS chapter:

“We used an in-house pipeline based on eCAVIAR⁷² to evaluate colocalization with GWAS summary statistics. The pipeline uses SuSiE-finemapped CSs as inputs, and calculates a causal posterior probability (CLPP) that the same locus is causal in both studies. CLPP is defined as the sum of the products of SuSiE-finemapped posterior probabilities (PIP; x for phenotype 1, y for phenotype 2) for each variant i shared in credible sets of both phenotypes, such that for credible set k:

$$[(CLPP)]_k = \sum_{(i \in CS)} [(x_i * y_i)]$$

CLPP was considered significant if it was higher than 20 %. Another colocalization

metric, causal posterior agreement (CLPA), was devised as a metric independent of CS size. CLPA represents the agreement between the fine-mapping results in both studies, and is defined as the sum of minimum PIP of shared variants between CSs from phenotype 1 and 2. CLPA was considered significant if higher than 50 %. For gene expression, the GWAS summary statistics were derived from GTEx v8⁷³ and EBI⁷⁴. Phenotype summary statistics were derived from the FinnGen PheWeb.

- Several variants in Table 4 appear to have allele frequencies that are inconsistent with other tables. For instance, rs189411872 has frequency 1.3% in Table S4, but 66% in Table S3
 - o We thank Reviewer #3 for catching this, there appeared to be inconsistencies in Table S3 in particular. We have now reviewed and corrected the allele frequencies in all Tables.

- In Table 3, please include the posterior probability of causality for each variant to help quantify the likelihood that the variant is causal. It also appears that rs11557467 and rs2305479 are two missense variants in different genes but belong to the same credible set? Please clarify.
 - o This is a valid point. We have now added the posterior probability to Table 4 (previously Table 3), which offers some potential insight to the interpretation of causal variants. The legend for Table 4 also reads “**16 non-synonymous variants** in protein-coding genes...” (previously mislabeled “functional”). We have also added the following statement to discuss the variants rs11557467 and rs2305479 in the 17q21.1 locus CS (p. 12, lines 247–253):

“The *IL1RL1* and *ZPBP2* missense variants have been previously associated with Type 2 high childhood asthma⁶¹ and adult-onset asthma⁶², respectively. The Finnish-enriched rs144651842 variant is in the well-known asthma locus *IL4R*^{63,64}. The *SLC22A4* and *FUT2* variants are linked with Crohn’s disease^{65,66}. **The *GSDMB* variant (rs2305479-T) was part of the same credible set as the asthma-linked *ZPBP2* missense variant rs11557467-T with high LD (r = 95 %) and lower posterior probability (1.5 % vs 4.0 %).**”

The locus is now also referred to in the discussion as follows (Line 370, p.18):
“[...] with the 17q21 locus (*GSDMB/ZPBP2*) as well [...]”

- Page 7. “Lead variants of the 58 non-HLA GWS loci were associated with 2,861 endpoints in FinnGen”. The phrasing here is ambiguous. On first reading I thought this meant that the 58 loci showed significant associations with 2861 other phenotypes, but I think this number refers to the total number of phenotypes tested?
 - o We apologize for the unclear phrasing, we indeed meant number of phenotypes tested. After the revisions to our colocalization of phenotypes, the sentence now reads (p. 14, lines 293–295):

“**To evaluate shared impact to non-IURD phenotypes, we used eCAVIAR⁷² to evaluate colocalization of IURD loci and endpoints in the FinnGen PheWeb (2,861 endpoints).**”

- Table S4. MAF column should be renamed EAF since many frequencies are > 0.5

- We thank the reviewer for noticing this, and have amended accordingly.
- *Table S7. Are the betas with respect to the same ref/effect alleles given in Tables S1-S3? Please clarify*
 - Yes. To clarify, we have now added the ref/effect alleles also to the cross-trait lead variants tables (now S2–S5).
- *“SNP” is used throughout the paper even though many of the lead variants at significant loci are indels.*
 - We apologize for the misnomer, and have amended accordingly

REVIEWERS' COMMENTS:

Reviewer #1 (Remarks to the Author: Overall significance):

The revision that followed the reviewer’s advices for change have drastically improved the manuscript. It is clear now what is claimed and how robust it is. Overall this is a milestone in the genetics of inflammatory diseases.

Reviewer #1 (Remarks to the Author: Impact):

This phenotype associations are of large interest since this might attract the attention of researchers studying other inflammatory diseases.

Up to now, genetic study of these phenotypes were limited.

This manuscript could fit Nature communications.

It possibly could fit Nature genetics, and whereas the result are now clear and interesting, they are not the first for most of the diseases, it is not clear that they bring a novel way of thinking in the field.

Again I recognize and appreciate the large improvement in presentation and results.

Reviewer #1 (Remarks to the Author: Strength of the claims):

Following the changes in the presentation, it the results are convincing and strong.

Reviewer #1 (Remarks to the Author: Reproducibility):

Following the reviewer comments, has motivated to clearly present replication. Overall the replications are convincing.

Reviewer #2 (Remarks to the Author: Overall significance):

The authors have gone to considerable effort to perform the extra analyses suggested by myself and the other reviewers. This has definitely improved the quality and rigour of the findings. That said, the paper continues to be rather heavy going and confusing– it reads like an automated text summary of a genetic results database. I’d recommend the authors invest considerable time thinking about how to simplify

the language and flow of results to help guide the reader. For example, I don't think I've ever read a more confusing summary of eQTL results from a GWAS before. Tracking the changing number of significant loci is also problematic. The last sentence before the 'IURD distinct heritability' section states 32 loci. This number then changes to 58 at the beginning of the replication section – I assume this difference is due to individual loci being associated with more than one trait. The authors then turn their 'replication' cohort into part of the discovery analysis, yielding 41 loci which are then used in all downstream analyses. Given this, it'd be more robust, conservative and simplifying to just include UKBB in the discovery from the outset, rather than using it as a pseudo-replication set which isn't used to distinguish false positives from true effects. If you're going to report the number of new loci when UKBB is included in the discovery it seems odd to 'claim' those which fell below genome-wide significance in the expanded discovery. Here the best analysis will always be the one in the largest sample size.

Reviewer #2 (Remarks to the Author: Impact):

The potential impact of this paper is quite limited currently by its writing style. It looks to be on the border of Communications Biology and Nature Communications

Reviewer #2 (Remarks to the Author: Strength of the claims):

Mentioned above

Reviewer #2 (Remarks to the Author: Reproducibility):

Already mentioned.

Reviewer #3 (Remarks to the Author: Overall significance):

Thank you to the authors for their thoughtful responses. I have no further comments.

Response to reviewer comments

Reviewer #1 comments:

Reviewer #1 (Remarks to the Author: Overall significance):

The revision that followed the reviewer's advices for change have drastically improved the manuscript. It is clear now what is claimed and how robust it is. Overall this is a milestone in the genetics of inflammatory diseases.

Reviewer #1 (Remarks to the Author: Impact):

This phenotype associations are of large interest since this might attract the attention of researchers studying other inflammatory diseases. Up to now, genetic study of these phenotypes were limited. This manuscript could fit Nature communications. It possibly could fit Nature genetics, and whereas the result are now clear and interisting, they are not the first for most of the diseases, it is not clear

that they bring a novel way of thinking in the field. Again I recognize and appreciate the large improvement in presentation and results.

Reviewer #1 (Remarks to the Author: Strength of the claims):

Following the changes in the presentation, it the results are convincing and strong.

Reviewer #1 (Remarks to the Author: Reproducibility):

Following the reviewer comments, has motivated to clearly present replication. Overall the replications are convincing.

We thank Reviewer #1 for their great feedback, and kind and encouraging words, here and previously.

Reviewer #2 comments:

Reviewer #2 (Remarks to the Author: Overall significance):

The authors have gone to considerable effort to perform the extra analyses suggested by myself and the other reviewers. This has definitely improved the quality and rigour of the findings. That said, the paper continues to be rather heavy going and confusing– it reads like an automated text summary of a genetic results database. I'd recommend the authors invest considerable time thinking about how to simplify the language and flow of results to help guide the reader. For example, I don't think I've ever read a more confusing summary of eQTL results from a GWAS before. Tracking the changing number of significant loci is also problematic. The last sentence before the 'IURD distinct heritability' section states 32 loci. This number then changes to 58 at the beginning of the replication section – I assume this difference is due to individual loci being associated with more than one trait. The authors then turn their 'replication' cohort into part of the discovery analysis, yielding 41 loci which are then used in all downstream analyses. Given this, it'd be more robust, conservative and simplifying to just include UKBB in the discovery from the outset, rather than using it as a pseudo-replication set which isn't used to distinguish false positives from true effects. If you're going to report the number of new loci when UKBB is included in the discovery it seems odd to 'claim' those which fell below genome-wide significance in the expanded discovery. Here the best analysis will always be the one in the largest sample size.

Reviewer #2 (Remarks to the Author: Impact):

The potential impact of this paper is quite limited currently by its writing style. It looks to be on the border of Communications Biology and Nature Communications

Reviewer #2 (Remarks to the Author: Strength of the claims):

Mentioned above

Reviewer #2 (Remarks to the Author: Reproducibility):

Already mentioned.

We thank Reviewer #2 for their feedback, and allowing us to improve our manuscript further.

I'd recommend the authors invest considerable time thinking about how to simplify the language and flow of results to help guide the reader.

We have now edited the Results section aiming to clarify the flow of the text. We hope that these edits have improved the readability of the manuscript.

For example, I don't think I've ever read a more confusing summary of eQTL results from a GWAS before.

Also this section has now been edited, e.g. aiming to clarify why specifically the results from cells involved in immune mechanisms are highlighted. We hope that these edits have clarified the section.

The last sentence before the 'IURD distinct heritability' section states 32 loci. This number then changes to 58 at the beginning of the replication section – I assume this difference is due to individual loci being associated with more than one trait

We agree with the reviewer that following the number of loci is too complicated. To simplify the flow, we have omitted to talk about the 58 loci. Instead, we discuss the number of loci that have been identified in each section and then just sum up the total number of loci, which is 41. We hope this clarifies

. Here the best analysis will always be the one in the largest sample size.

We very much agree with the reviewer that the largest sample is the best analysis. The challenge in this study comes from two aspects: 1) when large biobank studies are meta-analyzed, harmonizing phenotypes is always challenging. Some of the phenotypes are quite nicely concordant between UKBB and FinnGen, whereas some are not, often due to differences in the medical practice between countries. 2) In case of FinnGen we also have situations where Finnish enriched variants, that are very rare in other populations, are practically impossible to replicate in a traditional sense as those variants cannot be detected. This, obviously, doesn't mean that they would be false negatives. These are self-evident points, obviously very clear also to this reviewer, but we hope that some of the edits we have added, now clarify these two points.

Reviewer #3 comments:

Thank you to the authors for their thoughtful responses. I have no further comments.

We thank Reviewer #3 for their great feedback, here and previously.